# Random Projections and Sampling Algorithms for Clustering of High-Dimensional Polygonal Curves

**Stefan Meintrup**
Faculty of Computer Science
TU Dortmund University
Dortmund, Germany
stefan.meintrup@tu-dortmund.de

**Alexander Munteanu**
Dortmund Data Science Center
TU Dortmund University
Dortmund, Germany
alexander.munteanu@tu-dortmund.de

**Dennis Rohde**
Chair of Efficient Algorithms and Complexity Theory
TU Dortmund University
Dortmund, Germany
dennis.rohde@tu-dortmund.de

## Abstract

We study the $k$-median clustering problem for high-dimensional polygonal curves with finite but unbounded number of vertices. We tackle the computational issue that arises from the high number of dimensions by defining a Johnson-Lindenstrauss projection for polygonal curves. We analyze the resulting error in terms of the Fréchet distance, which is a tractable and natural dissimilarity measure for curves. Our clustering algorithms achieve sublinear dependency on the number of input curves via subsampling. Also, we show that the Fréchet distance can not be approximated within any factor of less than $\sqrt{2}$ by probabilistically reducing the dependency on the number of vertices of the curves. As a consequence we provide a fast, CUDA-parallelized version of the Alt and Godau algorithm for computing the Fréchet distance and use it to evaluate our results empirically.

## 1 Introduction

Time-series are sequences of measurements taken at certain instants of time. They arise in numerous applications, e.g., in the physical, geo-spatial, technical or financial domains (Zhang et al., 2007; Chapados and Bengio, 2008; Zimmer et al., 2018). Often there are multiple measurements per time instant, e.g., when there are numerous synchronized sensors. While the analysis of time-series is a well-studied topic, cf. Hamilton (1994); Liao (2005); Aghabozorgi et al. (2015), there are only few approaches that take high-dimensional multivariate time-series into account. In this work we build upon Driemel et al. (2016), who developed the first $(1 + \varepsilon)$-approximation algorithms for clustering univariate time-series under the Fréchet distance. Their idea is that –due to environmental circumstances– time-series often have heterogeneous lengths and their measurements are taken with different time-intervals in between. Thus, common approaches, where univariate time-series are represented by a point in a high-dimensional space, each dimension corresponding to one instant of time, become hard or even impossible to apply. Additionally, when these time-intervals differ substantially, depending on the sampling rates, continuous distance measures perform much better than discrete ones. This is due to the fact that they are inherently independent of the sampling rates: by interpreting a sequence of measurements as the vertices of a polygonal curve, those induce a linear interpolation between every two consecutive measurements. We extend this further to the multivariate case. When synchronized sensors are available, multiple univariate time-series are

interpreted as a high-dimensional polygonal curve, i.e., the number of dimensions equals the number of simultaneously measured attributes and the number of vertices equals the number of measurements.

We focus on big data with large number of curves $n$ and specifically on a large number of dimensions, say $d \in \Omega(n)$ as well as a high complexity of the curves, i.e., the number of their vertices is bounded by, say $m \in \Omega(n)$ each. This setting rules out the possibility of using sum-based continuous similarity measures like the continuous dynamic time warping distance, cf. Efrat et al. (2007). For this measure there is only one tractable algorithm, which is strongly related to paths on a two-dimensional manifold. Unfortunately, it also restricts to polygonal curves in $\mathbb{R}^2$. In contrast, the Alt and Godau algorithm (Alt and Godau (1995)) for computing the Fréchet distance works for any number of dimensions. The Fréchet distance intuitively measures the maximum distance one must traverse, when continuously and monotonously walking along two curves under an optimal speed adjustment, which is a suitable setting for comparing time-series in most cases. The Alt and Godau algorithm has running-time $\mathcal{O}(d \cdot m^2 \log(m))$, so we still end up with a worst-case running-time super-cubic in the number of input curves, in our setting. Unfortunately, it is impossible to reduce the complexity of the curves deterministically such that the Fréchet distance is preserved up to any multiplicative, which we prove in Theorem 14. Also it is not possible to reduce the complexity of the curves probabilistically such that the Fréchet distance is preserved up to any multiplicative less than $\sqrt{2}$, which we prove in Theorem 15. We tackle this issue by parallelizing the Alt and Godau algorithm via CUDA-enabled GPUs and thus preserve the original distance.

The main part of our work focuses on dimension reduction. SVD-based feature-selection approaches are common in practice, cf. Billsus and Pazzani (1998); Hong (1991). Unfortunately, these work poorly for polygonal curves, which we assess experimentally. Instead, we focus on Gaussian random projections via the seminal Johnson-Lindenstrauss Lemma (Johnson and Lindenstrauss, 1984) which perform much better. Explicit error-guarantees for discrete dissimilarity measures, like dynamic time warping or the discrete Fréchet distance, are immediate from the approximation of a finite number of Euclidean distances. But if we restrict to these measures, we loose the aforementioned linear interpolation which is not desirable in practice.

We thus study how the error of the Johnson-Lindenstrauss embedding propagates in the continuous case. In our theoretical analysis we show the first explicit error bound for the continuous Fréchet distance by extending the Johnson-Lindenstrauss embedding to polygonal curves. We project the vertices of the curve down from $d$ to $\mathcal{O}(\varepsilon^{-2}\log(nm))$ dimensions and re-connect their images in the low-dimensional space in the given order. The error is bounded by an $\varepsilon$-fraction relative to the Fréchet distance and to the length of the largest edge of the input curves, which we prove in Theorem 8. This gives a combined multiplicative and additive approximation guarantee, similar to the *lightweight coresets* of Bachem et al. (2018). All in all, we reduce the running-time of one Fréchet distance computation to $\mathcal{O}(\varepsilon^{-2} \frac{m^2}{\#cc} \log(m) \log(nm))$, where $\#cc$ is the number of CUDA cores available. We analyze various data sets. Our experiments show promising results concerning the approximation of the Fréchet distance under the Johnson-Lindenstrauss embedding and a massive improvement in terms of running-time.

Just as Driemel et al. (2016), we study median clustering. Since the median is a measure of central tendency that is robust when up to half of the data is arbitrarily corrupted, it is particularly useful for providing a summary of the massive data set. To the best of our knowledge, there is no tractable algorithm to compute an exact median polygonal curve. Thus, we restrict the search space of feasible solutions to the input. This problem, known as the discrete median, has a polynomial-time exhaustive-search algorithm: calculate the cost of each possible curve by summing over all other input curves. In our setting, this is prohibitive since it takes $\mathcal{O}(n^2)$ distance computations. Therefore, we propose and analyze a sampling-scheme for the discrete 1-median under the Fréchet distance when the number of input curves is also high. In Theorem 11 we show that a sample of constant size already yields a $(2 + \varepsilon)$-approximation in the worst case. Under reasonable assumptions on the distribution of the data, the same algorithm yields a $(1 + \varepsilon)$-approximation, which we prove in Theorem 12. To this end we introduce a natural parameter that quantifies the fraction of outliers as a function of the input, setting this approach in the light of *beyond worst-case analysis*, cf. Roughgarden (2019). The number of samples needed depends on this parameter and is almost always constant unless the fraction of outliers tends to $1/2$ at a high rate, depending on $n$. If those assumptions hold, we meet the requirements to apply Theorem 1.1 from Ackermann et al. (2010) and thus obtain a $k$-median $(1 + \varepsilon)$-approximation algorithm for the Fréchet distance that uses $n \cdot 2^{\mathcal{O}(\frac{k}{\varepsilon^2} + \log(\frac{k}{\varepsilon^3}))}$ distance computations.

Finally, we note that our techniques do not only apply to multivariate time-series, but to high-dimensional polygonal curves in general and thus may be valuable to the communities of computational geometry as well as the field of machine learning.

**Our contributions** We advance the study of clustering high-dimensional polygonal curves under the Fréchet distance both, in theory and in practice. Specifically,

1) we show an extension of the Gaussian random projections of Johnson-Lindenstrauss to polygonal curves and provide rigorous bounds on the distortion of their continuous Fréchet distance,

2) we provide sublinear sampling algorithms for the 1-median clustering of time series resp. polygonal curves under the Fréchet distance that can be extended (under natural assumptions) to a $k$-median $(1 + \varepsilon)$-approximation,

3) we prove lower bounds for reducing the curves complexity,

4) we provide a highly efficient CUDA-parallelized implementation of the algorithm by Alt and Godau (1995) for computing the Fréchet distance,

5) and we evaluate the proposed methods on benchmark and real-world data.

## 1.1 Related work

**Clustering under the Fréchet distance** Driemel et al. (2016) developed the first $k$-center and $k$-median clustering algorithms for one-dimensional polygonal curves under the Fréchet distance, which provably achieve an approximation factor of $(1 + \varepsilon)$. The resulting centers are curves from a discretized family of simplified curves, whose complexity is parameterized by a parameter $\ell$. Their algorithms have near-linear running-time in the input size for constant $\varepsilon$, $k$ and $\ell$ but are exponential in the latter quantities. The first extension of $k$-center to higher dimensional curves was done in Buchin et al. (2019a). In that paper, however it was shown that there is no polynomial-time approximation scheme unless P=NP. In the case of the discrete Fréchet distance on two-dimensional curves, the hardness of approximation within a factor close to 2.598 was established even for $k = 1$. Finally, Gonzalez' algorithm yields a 3-approximation in any number of dimensions. Even more recently Buchin et al. (2019b) showed that the $k$-median problem is also NP-hard for $k = 1$ and improved upon the aforementioned $(1+\varepsilon)$-approximations. Open problems thus include dimensionality reduction for high-dimensional curves and practical algorithms that do not depend exponentially on the parameters.

**Algorithm engineering for the Fréchet distance** Bringmann et al. (2019) describe an improved version of one of the best algorithms that was developed by the participants of the GIS Cup 2017. The goal of the cup was to answer Fréchet queries as fast as possible, i.e., given a set of curves $T$, a query curve $q$ and a positive real $r$, return all curves from $T$ that are within distance $r$ to $q$. Roughly speaking, all top algorithms (see also Baldus and Bringmann (2018); Buchin et al. (2017); Dütsch and Vahrenhold (2017)) utilized heuristics to filter out all $\tau \in T$, that are certainly within distance $r$ to $q$ or certainly not. In the best case, the common algorithm by Alt and Godau only served as a relapse option when no clear decision could be found in advance. Since the heuristics mostly have sublinear running-time, the Fréchet distance computation is speed up massively in the average case. The Alt and Godau algorithm is also improved by simplifying the resulting free-space diagram.

**Random projections for problems in computational geometry** Random projections have several applications as embedding techniques in computational geometry. One of the most influential work was Agarwal et al. (2013) who applied the Johnson-Lindenstrauss embedding, among others, to surfaces and curves for the sake of tracking moving points. Only recently Driemel and Krivosija (2018) studied the first probabilistic embeddings of the Fréchet distance by projecting the curves on a random line. Another work that inspired our dimensionality reduction approach is due to Sheehy (2014). He noticed that a Johnson-Lindenstrauss embedding of points yields an embedding for their entire convex hull with additive error. Our results are in line with a recent lower bound of $\Omega(n)$ for sketching, i.e., compressing the strongly related Dynamic Time Warping distance of sequences via linear embeddings, due to Braverman et al. (2019).

**Beyond-worst-case and relaxations** A common assumption is that *"Clustering is difficult only when it does not matter"* (Daniely et al., 2012). Similarly, it has been noted for many other problems that while being particularly hard to solve in the worst-case, they are relatively simple to solve for *typical* or slightly perturbed inputs. Beyond-worst-case-analysis tries to parametrize the notion of *typical*

and to derive better bounds in terms of this parameter assuming its value is small. See Munteanu et al. (2018) for a recent contribution in machine learning. These assumptions are usually weaker than the norm in statistical machine learning which is closer to average-case analysis, for example when data points are modeled as i.i.d. samples from some distribution. See (Roughgarden, 2019) for an extensive overview and more details. Another complementary recent approach is weakening the usual multiplicative error guarantees by an additional additive error term in favor of a computational speedup. Those relaxations still perform competitively well in practice, cf. (Bachem et al., 2018).

## 2 Dimension Reduction for Polygonal Curves

We begin with the basic definitions, all proofs can be found in Appendix A in the supplement. Polygonal curves are composed of line segments, which we define as follows.

**Definition 1** (line segment). *A line segment between two points $p_1, p_2 \in \mathbb{R}^d$, denoted by $\overline{p_1 p_2}$, is the set of points $\{(1 - \lambda)p_1 + \lambda p_2 \mid \lambda \in [0, 1]\}$. For $\lambda \in [0, 1]$ we denote by $\mathrm{lp}\left(\overline{p_1 p_2}, \lambda\right)$ the point $(1 - \lambda)p_1 + \lambda p_2$, lying on $\overline{p_1 p_2}$.*

We next define polygonal curves. Thereby we need an exact parametrization of the points on the individual line segments to express any point on the curve in terms of its segments vertices. This unusually complicates the definition but simplifies the notation and will later be needed in the context of Johnson-Lindenstrauss embeddings.

**Definition 2** (polygonal curve). *A parameterized curve is a continuous mapping $\tau\colon [0, 1] \to \mathbb{R}^d$. Let $\mathcal{H}$ be the set of all continuous, injective and non-decreasing functions $h\colon [0, 1] \to [0, 1]$ with $h(0) = 0$ and $h(1) = 1$, which we call reparameterizations.*

*A curve $\tau$ is polygonal, if there exist $h \in \mathcal{H}$, $v_1, \ldots, v_m \in \mathbb{R}^d$, no three consecutive on a line, called $\tau$'s vertices and $t_1, \ldots, t_m \in [0, 1]$ with $t_1 < \cdots < t_m$, $t_1 = 0$ and $t_m = 1$, called $\tau$'s instants, such that*

$$\tau(h(t)) = \begin{cases} \mathrm{lp}\left(\overline{v_1 v_2}, \frac{h(t) - t_1}{t_2 - t_1}\right), & \text{if } h(t) \in [0, t_2) \\ \vdots \\ \mathrm{lp}\left(\overline{v_{m-1} v_m}, \frac{h(t) - t_{m-1}}{t_m - t_{m-1}}\right), & \text{if } h(t) \in [t_{m-1}, 1] \end{cases}.$$

In the following we will assume that $h$ is the identity function, because the Fréchet distance, which is subsequently defined, is invariant under reparameterizations. We only need $h$ to keep our definition general. Further, we call $m$ the complexity of $\tau$, denoted by $|\tau|$. We are now ready to define the (continuous) Fréchet distance.

**Definition 3** (continuous Fréchet distance). *The Fréchet distance between polygonal curves $\tau$ and $\sigma$ is defined as $d_F(\tau, \sigma) := \inf_{h \in \mathcal{H}} \max_{t \in [0,1]} \|\tau(t) - \sigma(h(t))\|$, where $\|\cdot\|$ is the Euclidean norm.*

We next give a basic defintion of the seminal Johnson-Lindenstrauss embedding result, cf. Johnson and Lindenstrauss (1984). Specifically, they showed that a properly rescaled Gaussian matrix mapping from $d$ to $d' \in O(\varepsilon^{-2} \log n)$ dimensions satisfies the following definition with positive constant probability.

**Definition 4** ($(1 \pm \varepsilon)$-Johnson-Lindenstrauss embedding). *Given a set $P \subset \mathbb{R}^d$ of points, a function $f\colon \mathbb{R}^d \to \mathbb{R}^{d'}$ is a $(1 \pm \varepsilon)$-Johnson-Lindenstrauss embedding for $P$, if it holds that*

$$\forall p, q \in P : (1 - \varepsilon)\|p - q\| \leq \|f(p) - f(q)\| \leq (1 + \varepsilon)\|p - q\|,$$

*with constant probability at least $\rho \in (0, 1]$ over the random construction of $f$.*

In Definition 5 we extend the mapping $f$ from Definition 4 to polygonal curves by applying it to the vertices of the curves and re-connecting their images in the given order.

**Definition 5** ($(1 \pm \varepsilon)$-Johnson-Lindenstrauss embedding for polygonal curves). *Let $\tau$ be a polygonal curve, $t_1, \ldots, t_m$ be its instants and $v_1, \ldots, v_m$ be its vertices. Let $f$ be a $(1 \pm \varepsilon)$-Johnson-Lindenstrauss embedding for $\{v_1, \ldots, v_m\}$. By $F(\tau)$ we define the $(1 \pm \varepsilon)$-Johnson-Lindenstrauss*

*embedding of $\tau$ as follows:*

$$F(\tau)(t) := \begin{cases} \mathrm{lp}\left(\overline{f(v_1)f(v_2)}, \frac{t-t_1}{t_2-t_1}\right), & \text{if } t \in [0, t_2) \\ \vdots & \\ \mathrm{lp}\left(\overline{f(v_{m-1})f(v_m)}, \frac{t-t_{m-1}}{t_m-t_{m-1}}\right), & \text{if } t \in [t_{m-1}, 1] \end{cases}.$$

*For a set $T := \{\tau_1, \ldots, \tau_n\}$ of polygonal curves we define $F(T) := \{F(\tau) \mid \tau \in T\}$ and require the function $f$ to be a $(1 \pm \varepsilon)$-Johnson-Lindenstrauss embedding for the set of* all *vertices of* all *$\tau \in T$.*

We next give an explicit bound on the distortion of the Fréchet distance when the map of Definition 5 is applied to the input curves. Note that the previously mentioned approach by Sheehy (2014) for the convex hull of points is not directly applicable since two curves might be drawn apart from each other making the error arbitrary large. Our additive error will depend only on the length of line segments between consecutive points of a curve, which is usually bounded.

We first express the distance between two points on two distinct line segments using their relative positions on the respective line segment.

**Proposition 6.** *Let $s_1 := \overline{p_1p_2}$ and $s_2 := \overline{q_1q_2}$ be line segments between two points $p_1 := (p_{1,1}, \ldots, p_{1,d}), p_2 := (p_{2,1}, \ldots, p_{2,d}) \in \mathbb{R}^d$, respective $q_1 := (q_{1,1}, \ldots, q_{1,d}), q_2 := (q_{2,1}, \ldots, q_{2,d}) \in \mathbb{R}^d$. For any $\lambda_p, \lambda_q \in [0, 1]$ and $p := \mathrm{lp}(\overline{p_1p_2}, \lambda_p)$ lying on $s_1$, as well as $q := \mathrm{lp}(\overline{q_1q_2}, \lambda_q)$ lying on $s_2$, it holds that*

$$\begin{aligned} \|p-q\|^2 = & -(\lambda_p - \lambda_p^2)\|p_1 - p_2\|^2 - (\lambda_q - \lambda_q^2)\|q_1 - q_2\|^2 + (1 - \lambda_p - \lambda_q + \lambda_p\lambda_q)\|p_1 - q_1\|^2 \\ & + (\lambda_q - \lambda_p\lambda_q)\|p_1 - q_2\|^2 + (\lambda_p - \lambda_p\lambda_q)\|p_2 - q_1\|^2 + \lambda_p\lambda_q\|p_2 - q_2\|^2. \end{aligned}$$

Proposition 6 can be proven using the law of cosines, the geometric and algebraic definition of the dot product and tedious algebraic manipulations.

Using Proposition 6, our calculation yields an explicit error-bound when applying Definition 5 to both line-segments. This is formalized in Lemma 7.

**Lemma 7.** *Let $P := \{p_1, \ldots, p_n\} \subset \mathbb{R}^d$ be a set of points and $f$ be a $(1 \pm \varepsilon)$-Johnson-Lindenstrauss embedding for $P$. Let $p_1, p_2, q_1, q_2 \in P$, for arbitrary $\lambda_p, \lambda_q \in [0, 1]$ and $p := \mathrm{lp}(\overline{p_1p_2}, \lambda_p)$, $p' := \mathrm{lp}\left(\overline{f(p_1)f(p_2)}, \lambda_p\right)$, as well as $q := \mathrm{lp}(\overline{q_1q_2}, \lambda_q)$, $q' := \mathrm{lp}\left(\overline{f(q_1)f(q_2)}, \lambda_q\right)$ it holds that*

$$(1-\varepsilon)^2\|p-q\|^2 - \varepsilon(\|p_1 - p_2\|^2 + \|q_1 - q_2\|^2) \le \|p'-q'\|^2 \le (1+\varepsilon)^2\|p-q\|^2 + \varepsilon(\|p_1 - p_2\|^2 + \|q_1 - q_2\|^2)$$

*is satisfied with probability at least $\rho \in (0, 1]$ over the random construction of $f$.*

This finally yields our main theorem which states the desired error guarantee for the Fréchet distance of a set of polygonal curves.

**Theorem 8.** *Let $T := \{\tau_1, \ldots, \tau_n\}$ be a set of polygonal curves and for $\tau \in T$ let $\alpha(\tau)$ denote the maximum distance of two consecutive vertices of $\tau$. Furher, for $\tau, \sigma \in T$ let $\alpha(\tau, \sigma) := \max\{\alpha(\tau), \alpha(\sigma)\}$. Now, let $F$ be a $(1 \pm \varepsilon)$-Johnson-Lindenstrauss embedding for $T$. With constant probability at least $\rho \in (0, 1]$ it holds for all $\tau, \sigma \in T$ that*

$$\sqrt{(1-\varepsilon)^2 d_F^2(\tau, \sigma) - 2\varepsilon\alpha(\tau, \sigma)^2} \le d_F(F(\tau), F(\sigma)) \le \sqrt{(1+\varepsilon)^2 d_F^2(\tau, \sigma) + 2\varepsilon\alpha(\tau, \sigma)^2},$$

*where the exact value for $\rho$ stems from the technique used for obtaining $f$.*

Let us first note that these bounds tend to $d_F(\tau, \sigma)$ as $\varepsilon$ tends to 0. The multiplicative error bounds are similar to $\varepsilon$-coresets which are popular data reduction techniques in clustering, cf. (Feldman et al., 2010, 2013; Sohler and Woodruff, 2018). The additional additive error is in line with the relaxation given by *lightweight coresets* (Bachem et al., 2018).

We believe that the additive error is necessary. Consider the following two polygonal curves in $\mathbb{R}^d$, for $d \ge 3$. Let $\alpha \in \mathbb{R}_{>0}$ be arbitrary. The first curve is $p := \overline{p_1p_2}$ with $p_1 := (0, \ldots, 0)$ and $p_2 := (\alpha, 0, \ldots, 0)$. The second curve $q$ has vertices $q_1 := (0, 1, 0, \ldots, 0)$, $q_2 := (\frac{\alpha}{2}, 2, 1, 0, \ldots, 0)$ and $q_3 := (\alpha, 1, 0, \ldots, 0)$. It's edges are $\overline{q_1q_2}$ and $\overline{q_2q_3}$. Clearly, we have $\|p_1 - p_2\| = \alpha = \|q_1 - q_3\|$,

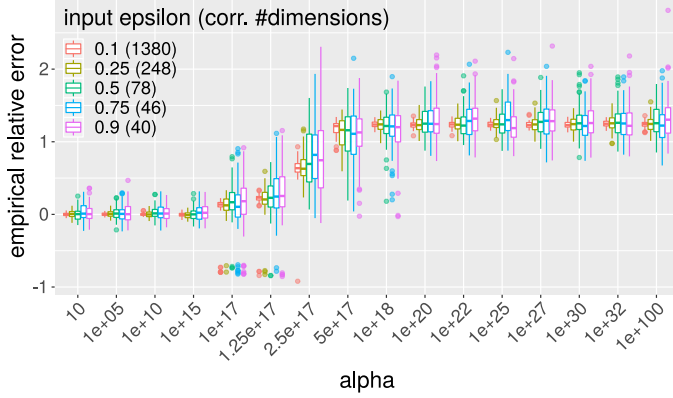

Figure 1: Empirical relative error in terms of the distortion of the Fréchet distance between the curves $p$ and $q$. It can be observed that the distortion depends on the value $\alpha$, which determines the curves lengths, but *not* their Fréchet distance. Note that an empirical relative error above 1 means that not even a 2-approximation of the distance was achieved. However, this only happens for line segments of length larger than $10^{15}$.

$\|p_1 - q_1\| = 1 = \|p_2 - q_3\|$ and $\|p_1 - q_2\| = (\frac{\alpha^2}{4} + 5)^{1/2} = \|p_2 - q_2\|$, as well as $\|q_1 - q_2\| = (\frac{\alpha^2}{4} + 2)^{1/2} = \|q_2 - q_3\|$. Also note that $d_F(p,q) = \sqrt{5}$, a constant that does not depend on $\alpha$. The pairwise distances among the points will be distorted by at most $(1 \pm \varepsilon)$. Now the embedding has its mass concentrated in the interval $(1 \pm \varepsilon)$ but inspecting the concentration inequalities most of this mass is between $(1 \pm \frac{\varepsilon}{c})$ and $(1 \pm \varepsilon)$ for large $c$. Thus, with reasonably large probability the error on $q_2$ will depend on $\frac{\varepsilon\alpha}{c}$ which is additive since $\alpha$ is unrelated to the original Fréchet distance.

We assess the distortion of the Fréchet distance between $p$ and $q$ experimentally. We use the target dimension of the proof in (Dasgupta and Gupta, 2003) and all combinations of five choices for $\varepsilon$, as well as sixteen choices for $\alpha$, we conduct one experiment with one hundred repetitions. The results are depicted in Fig. 1.

## 3   Median Clustering under the Fréchet Distance

We study the $k$-median problem. As discussed before, we restrict the centers to subsets of the input.

**Definition 9** (discrete median clustering). *Given a set of $T$ of polygonal curves, the $k$-median clustering problem is to find a set $C \subseteq T$ of $k$ centers such that the sum of the distances from the curves in $T$ to the closest center in $C$ is minimized.*

At first, we restrict to $k = 1$. Instead of exhaustively trying out all curves as possible median, thus computing all pairwise distances among the input curves, we aim to find a small *candidate* set of possible medians and another small *witness* set which serves as a proxy to sum over. We will use the following theorem of Indyk (2000) to bound the number of required witnesses, given a set of candidates of certain size.

**Theorem 10.** *(Indyk, 2000, Theorem 31) Let $\varepsilon \in (0,1]$ be a constant and $T$ be a set of polygonal curves. Further let $W$ be a non-empty uniform sample from $T$. For $\tau, \sigma \in T$ with $\sum_{\tau' \in T} d_F(\tau, \tau') > (1 + \varepsilon) \sum_{\tau' \in T} d_F(\sigma, \tau')$ it holds that $\Pr[\sum_{\tau' \in W} d_F(\tau, \tau') \leq \sum_{\tau' \in W} d_F(\sigma, \tau')] < \exp\left(-\varepsilon^2 |W|/64\right)$.*

Using only this theorem, we still have to cope with all $n$ input curves as candidates. In what follows, we reduce this to a constant size sample of the input. Without assumptions on the input, by standard probabilistic arguments and the triangle-inequality, we obtain a $(2 + \varepsilon)$-approximation.

**Theorem 11.** *Given constants $\varepsilon, \delta \in (0, 1/2)$ and a non-empty set $T$ of polygonal curves, we can use a uniform sample $S := \{s_1, \ldots, s_{\ell_S}\}$ of cardinality $\mathcal{O}\left(\ln(1/\delta)/\varepsilon\right)$ of candidates and a uniform sample $W := \{w_1, \ldots, w_{\ell_W}\}$ of cardinality $\mathcal{O}\left(\ln(\ell_S/\delta)/\varepsilon^2\right)$ of witnesses, to obtain a $(2 + \varepsilon)$-approximate 1-median $c_S \in S$ with probability at least $1 - \delta$.*

Under natural assumptions, setting our analysis in the Beyond-Worst-Case regime (Roughgarden, 2019), we can even get a $(1 + \varepsilon)$-approximation on a subsample of sublinear size. The high-level idea behind is that the curves are usually not equidistant to an optimal median. Relative to the average cost, there will be some outliers, some curves at medium distance and also some curves very close to an optimal median. Now if there are quite a good number of outliers, but also not too many, they

make up a good share of the total cost. This implies that the number of curves at medium distance is bounded by a constant fraction of the curves. Finally this implies that the number of curves that are close to an optimal median, is not too small such that a small sample will include at least one of them with constant probability.

**Theorem 12.** *Let $\varepsilon, \delta \in (0, 1/2)$ be constants, and $T$ be a non-empty set of polygonal curves with at least $(1 - \varepsilon)\gamma(T)n$ outliers, for $0 < \gamma(T) < 1/2$. We can use a uniform sample $S := \{s_1, \ldots, s_{\ell_S}\}$ of cardinality $\ell_S = \mathcal{O}\left(\frac{\ln(1/\delta)}{1/2 - \gamma(T)}\right)$ of candidates and a uniform sample $W := \{w_1, \ldots, w_{\ell_W}\}$ of cardinality $\mathcal{O}\left(\ln(\ell_S/\delta)/\varepsilon^2\right)$ of witnesses, to obtain a $(1 + \varepsilon)$-approximate 1-median $c_S \in S$ with probability at least $1 - \delta$.*

Note in particular, that the samples in Theorem 12 still have constant size unless the fraction of outliers $\gamma(T)$ tends arbitrarily close to $1/2$ depending on $|T| = n$. In this case the usual notion of an outlier is not met for two reasons: first, more than a quarter of the curves would be considered outliers, and second their distance to an optimal median is not much larger than the medium curves implying that basically all curves are in a narrow annulus around the average distance. Both observations make the notion of outliers highly questionable. The details are in the proof and Fig. 5, which can be found in the supplement. Note that in practice it is neither necessary nor desirable to compute $\gamma(T)$. Instead, one should set $\gamma(T) = 1/2 - 1/c$, for a large enough constant $c$. Now, if our assumptions on the input hold, $d_F$ has the $[\varepsilon, \delta]$-sampling property from Ackermann et al. (2010) and we can apply their Theorem 1.1, yielding the following corollary:

**Corollary 13.** *Under the assumptions of Theorem 12, there exists an algorithm for the discrete $k$-median under the Fréchet distance that, given a set of $n$ polygonal curves and $\varepsilon \in (0, 1)$, returns with positive constant probability a $(1 + \varepsilon)$-approximation using only $n \cdot 2^{O(k \cdot (|S| + |W|) \log(\frac{k}{\varepsilon} \cdot (|S| + |W|)))}$ distance computations, where $S$ is the candidate sample and $W$ is the witness sample.*

## 4 Complexity Reduction for Polygonal Curves

We study the space complexity of compressing polygonal curves such that their complexity, i.e., their number of vertices, is reduced while their Fréchet distance is preserved. Recall that $m$ dominates the running-time of the Alt and Godau algorithm. Now, for reducing this dependence, the goal is to define a randomized function $S$ together with an estimation procedure $E$ so that for any polygonal curves $\tau, \sigma$, we take the compressed representations $S(\tau)$ and the estimation procedure satisfies with constant probability $d_F(\tau, \sigma) \leq E(S(\tau), \sigma) \leq \eta \cdot d(\tau, \sigma)$ for some approximation factor $\eta$, cf. Braverman et al. (2019). The challenge is to bound the size of $S(\tau)$ depending on the complexity of $\tau$ in order to obtain an approximation factor of $\eta$.

We prove that the Fréchet distance can not be approximated up to any factor by reducing the complexity of the curves deterministically, even in one dimension. We achieve this result by reducing from the equality test communication problem, which requires a linear number of bits, cf. Wegener (2005).

**Theorem 14.** *Let $\tau, \sigma$ be polygonal curves in $\mathbb{R}^d$, for $d \geq 1$, with $m$ vertices each. Any deterministic data oblivious sketching function $S$ for which there exists a deterministic estimation function $E$ satisfying $d_F(\tau, \sigma) \leq E(S(\tau), \sigma) \leq \eta \cdot d_F(\tau, \sigma)$, for an arbitrary $\eta \in [1, \infty)$, uses $\Omega(m)$ bits to represent $S(\tau)$.*

Also, we prove that the Fréchet distance can not be approximated within any factor less than $\sqrt{2}$ by reducing the complexity of the curves probabilistically. We show this by reducing from the set disjointness communication problem, which also requires a linear number of bits for any randomized protocol succeeding with constant probability, cf. Håstad and Wigderson (2007).

**Theorem 15.** *Let $\tau, \sigma$ be polygonal curves in $\mathbb{R}^d$, for $d \geq 2$, with $m$ vertices each. Any randomized data oblivious sketching function $S$ for which there exists a randomized estimation function $E$ satisfying $d_F(\tau, \sigma) \leq E(S(\tau), \sigma) \leq \eta \cdot d_F(\tau, \sigma)$, for $\eta \in [1, \sqrt{2}]$, uses $\Omega(m)$ bits to represent $S(\tau)$.*

## 5 Experiments

The main practical motivation for our work is that any application utilizing the Fréchet distance suffers from its computational cost. In general, there are three parameters on which the running-time

Figure 2: (a): Distortion under the $(1 \pm \varepsilon)$-Johnson-Lindenstrauss embedding. The lateral axis shows the values for $\varepsilon$ plugged into the embedding (and the corresponding number of dimensions). The longitutdinal axis shows the empirical relative error. (b): Running-times of the algorithms, where sequential is an naïve implementation of the Alt and Godau algorithm, parallel is our CUDA-enabled variant and the suffix "_rp" means that the data was randomly projected before.

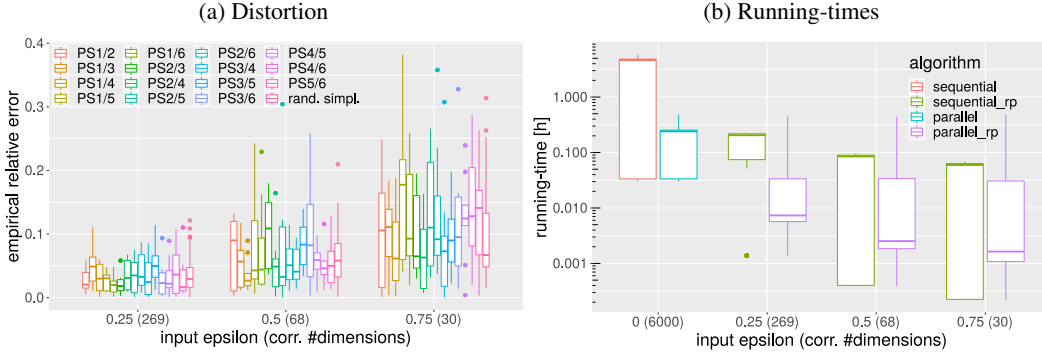

depends: the dimension of the ambient space $d$, the number of curves $n$ and their complexity $m$. We tackle the first utilizing our results from Section 2, i.e., the dimension reduction and the second by utilizing our results from Section 3, i.e., the sampling schemes. For the last, by Section 4 we would loose a factor of at least $\sqrt{2}$ and can not hope to design a $(1 + \varepsilon)$-approximation algorithm with subquadratic running-time in $m$. We thus decide to tackle the dependence of the Alt and Godau algorithm on $m$ by parallelization.[1] We now seek to answer:

**Q1** Does the random projection induce a reasonably small distortion on the Fréchet distance?

**Q2** What is the impact of our techniques on the running-time of the Fréchet distance computation?

**Q3** Do we obtain reasonable results combining the sampling scheme and the random projection?

**Q4** Does PCA lead to better results than random projections?

Before we answer these questions based on our experimental results, we describe our data sets, the modifications we applied to the Alt and Godau algorithm, and our setup. Also, note that we used the empirical constant of 2 for the experiments in this section, cf. Venkatasubramanian and Wang (2011). Therefore, we projected from $d$ to $d' = 2\varepsilon^{-2} \ln(nm)$ dimensions.

**Data sets** Our first data set was taken by monitoring a hydraulic test rig via multiple sensors (cf. Helwig et al. (2015)), including six pressure sensors PS1, . . . , PS6. In a total of 2205 test-cycles, each sensor measured 6000 values in each cycle. We chose to build six polygonal curves with 2205 vertices each in the 6000-dimensional Euclidean space. Also, for comparison we generate curves of equal complexity and ambient dimension by picking their vertices uniformly at random from a $d + 1$-simplex scaled by a large number, thus obtaining curves of high intrinsic dimension. For 1-median clustering we use weather simulation data (Lucas et al., 2015) from which we construct 2922 curves with 15 vertices each in 327-dimensional Euclidean space.

**Algorithm modifications** We decided to parallelize the Alt and Godau algorithm utilizing CUDA-enabled graphic cards. We improve the worst-case running-time of the algorithm from $\mathcal{O}(dm^2 \log(m))$ to $\mathcal{O}(d\frac{m^2}{\#cc} \log(m))$, where $\#cc$ is the number of available CUDA cores.

**Setup** We ran our experiments on a high perfomance linux cluster, which has twenty GPU nodes with two Intel Xeon E5-2640v4 CPUs, 64 GB of RAM and two Nvidia K40 GPUs each. This makes 2880 CUDA cores per card. To minimize interference, each experiment was run on an exclusive core and both GPUs, with 30 GB of RAM guaranteed. Each experiment was run ten times for each parametrization. Every experiment concerning the curves sampled from the simplex was even run one hundred times for each parameterization.

Figure 3: (a): Running-times and (b): deviations for the Fréchet 1-median sampling scheme. The lateral axis shows the values for $\varepsilon$ plugged into the sampling algorithm. The deviations are with respect to the optimal objective value. *epsilon rp* is the value for $\varepsilon$ that is plugged into the embedding.

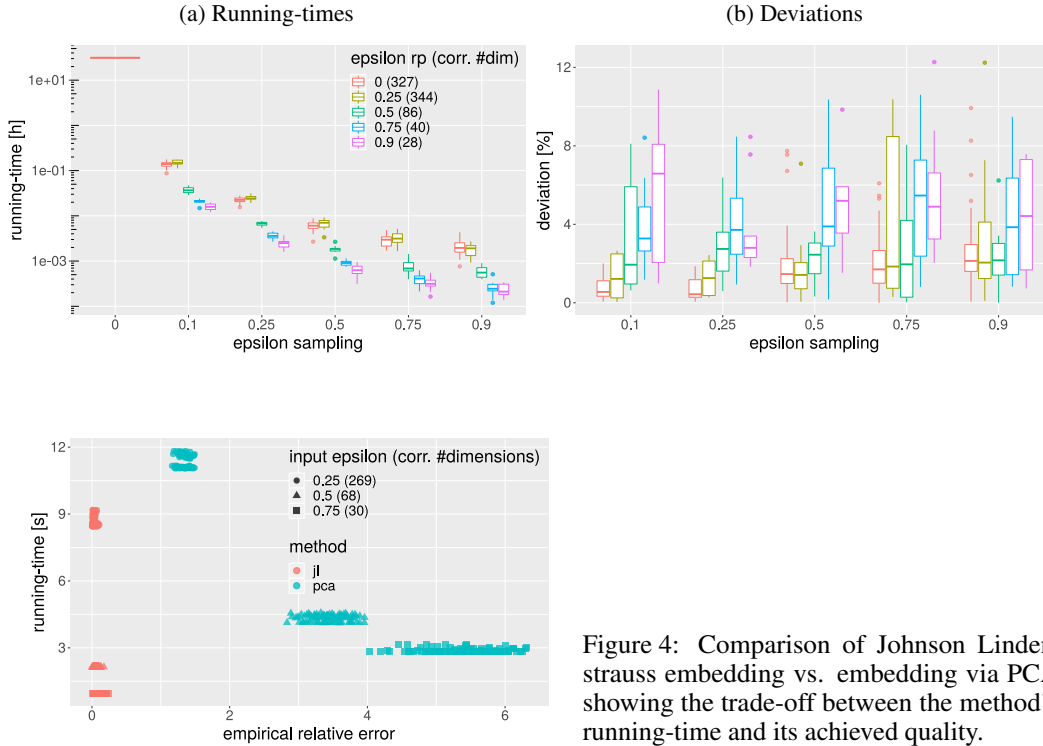

Figure 4: Comparison of Johnson Lindenstrauss embedding vs. embedding via PCA showing the trade-off between the method's running-time and its achieved quality.

**Q1** Concerning all data sets we can say that the distortion of the Fréchet distance after applying the Johnson-Lindenstrauss embedding is reasonably small. In Fig. 2(a) we depict the results of the Fréchet distance computations vs. the chosen values for $\varepsilon$. It can be observed that even for larger values of $\varepsilon$, the effective error never exceeds the given margin.

**Q2** In Fig. 2(b) we depict the running-times of the Alt and Godau algorithm under our measures. The results stem from the same experiments that lead to the values depicted in Fig. 2(a). The random projection and the parallelization speed up the computations by a factor of 10 each independently. Both together yield a speedup of factor 100. While the naïve implementation of the algorithm took about roughly three hours, we were able to lower the running-time to about 30 seconds on average.

**Q3** We conducted experiments on the weather simulation dataset. Fig. 3 shows that employing the subsampling schemes yields substantial improvements in terms of running-times while the approximation error remains robust to the choices of the approximation parameters "epsilon sampling" and "epsilon rp" plugged into the subsampling scheme and the embedding, respectively. This indicates that the approximation is indeed dependent on the data paramater $\gamma(T)$.

**Q4** In Fig. 4 we compare the Johnson-Lindenstrauss embedding for polygonal curves to PCA applied to the vertices in a similar fashion. Here, we only used the curves whose vertices were sampled from a $d + 1$-simplex to emphasize the impact of hard inputs on the distortion. We depict the methods running-time vs. distortion. It can be observed that for all choices of $\varepsilon$, the Johnson Lindenstrauss embedding performs much better in terms of distortion as well as running-time.

## Acknowledgments

We thank the anonymous reviewers for their valuable comments. This work was supported by the German Science Foundation (DFG), Collaborative Research Center SFB 876 "Providing Information by Resource-Constrained Analysis", project C4 and by the Dortmund Data Science Center (DoDSc).

## Footnotes

[1]Code available at `https://www.dennisrohde.work/rp4frechet-code`.

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
