[Supplementary Material 1]

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

## A Omitted material

*Proof of Proposition 6.* We have:

$$\|p - q\|^2 = \sum_{i=1}^{d}[(p_{1,i} - \lambda_p(p_{1,i} - p_{2,i})) - (q_{1,i} - \lambda_q(q_{1,i} - q_{2,i}))]^2 \tag{I}$$

$$= \sum_{i=1}^{d}[(p_{1,i} - \lambda_p(p_{1,i} - p_{2,i}))^2 - 2(p_{1,i} - \lambda_p(p_{1,i} - p_{2,i}))(q_{1,i} - \lambda_q(q_{1,i} - q_{2,i}))$$
$$+ (q_{1,i} - \lambda_q(q_{1,i} - q_{2,i}))^2] \tag{II}$$

$$= \sum_{i=1}^{d}[p_{1,i}^2 - 2\lambda_p p_{1,i}(p_{1,i} - p_{2,i}) + \lambda_p^2(p_{1,i} - p_{2,i})^2$$
$$- 2(p_{1,i} - \lambda_p(p_{1,i} - p_{2,i}))(q_{1,i} - \lambda_q(q_{1,i} - q_{2,i})) + q_{1,i}^2 - 2\lambda_q q_{1,i}(q_{1,i} - q_{2,i})$$
$$+ \lambda_q^2(q_{1,i} - q_{2,i})^2] \tag{III}$$

$$= \sum_{i=1}^{d}[p_{1,i}^2 - 2\lambda_p p_{1,i}^2 + 2\lambda_p p_{1,i}p_{2,i} + \lambda_p^2 p_{1,i}^2 - 2\lambda_p^2 p_{1,i}p_{2,i} + \lambda_p^2 p_{2,i}^2 - 2p_{1,i}q_{1,i}$$
$$+ 2\lambda_q p_{1,i}(q_{1,i} - q_{2,i}) + 2\lambda_p q_{1,i}(p_{1,i} - p_{2,i}) - 2\lambda_p\lambda_q(p_{1,i} - p_{2,i})(q_{1,i} - q_{2,i})$$
$$+ q_{1,i}^2 - 2\lambda_q q_{1,i}^2 + 2\lambda_q q_{1,i}q_{2,i} + \lambda_q^2 q_{1,i}^2 - 2\lambda_q^2 q_{1,i}q_{2,i} + \lambda_q^2 q_{2,i}^2] \tag{IV}$$

$$= \sum_{i=1}^{d}[(1 - 2\lambda_p + \lambda_p^2)p_{1,i}^2 + \lambda_p^2 p_{2,i}^2 + (1 - 2\lambda_q + \lambda_q^2)q_{1,i}^2 + \lambda_q^2 q_{2,i}^2$$
$$+ 2\lambda_p(1 - \lambda_p)p_{1,i}p_{2,i} - 2p_{1,i}q_{1,i} + 2\lambda_q p_{1,i}q_{1,i} - 2\lambda_p p_{1,i}q_{2,i} + 2\lambda_p p_{1,i}q_{1,i}$$
$$- 2\lambda_p p_{2,i}q_{1,i} - 2\lambda_p\lambda_q p_{1,i}q_{1,i} + 2\lambda_p\lambda_q p_{2,i}q_{1,i} + 2\lambda_p\lambda_q p_{1,i}q_{2,i}$$
$$- 2\lambda_p\lambda_q p_{2,i}q_{2,i} + 2\lambda_q(1 - \lambda_q)q_{1,i}q_{2,i}] \tag{V}$$

$$= (1 - \lambda_p)^2\|p_1\|^2 + \lambda_p^2\|p_2\|^2 + (1 - \lambda_q)^2\|q_1\|^2 + \lambda_q^2\|q_2\|^2$$
$$+ 2\lambda_p(1 - \lambda_p)\langle p_1, p_2\rangle + 2(\lambda_p + \lambda_q - \lambda_p\lambda_q - 1)\langle p_1, q_1\rangle$$
$$+ 2(\lambda_p\lambda_q - \lambda_q)\langle p_1, q_2\rangle + 2(\lambda_p\lambda_q - \lambda_p)\langle p_2, q_1\rangle - 2\lambda_p\lambda_q\langle p_2, q_2\rangle$$
$$+ 2\lambda_q(1 - \lambda_q)\langle q_1, q_2\rangle \tag{VI}$$

$$= (1 - \lambda_p)^2\|p_1\|^2 + \lambda_p^2\|p_2\|^2 + (1 - \lambda_q)^2\|q_1\|^2 + \lambda_q^2\|q_2\|^2$$
$$+ 2\lambda_p(1 - \lambda_p)\|p_1\|\|p_2\|\cos\sphericalangle(p_1, p_2)$$
$$+ 2(\lambda_p + \lambda_q - \lambda_p\lambda_q - 1)\|p_1\|\|q_1\|\cos\sphericalangle(p_1, q_1)$$
$$+ 2(\lambda_p\lambda_q - \lambda_q)\|p_1\|\|q_2\|\cos\sphericalangle(p_1, q_2)$$
$$+ 2(\lambda_p\lambda_q - \lambda_p)\|p_2\|\|q_1\|\cos\sphericalangle(p_2, q_1)$$
$$- 2\lambda_p\lambda_q\|p_2\|\|q_2\|\cos\sphericalangle(p_2, q_2) + 2\lambda_q(1 - \lambda_q)\|q_1\|\|q_2\|\cos\sphericalangle(q_1, q_2) \tag{VII}$$

$$= (1 - \lambda_p)^2\|p_1\|^2 + \lambda_p^2\|p_2\|^2 + (1 - \lambda_q)^2\|q_1\|^2 + \lambda_q^2\|q_2\|^2$$
$$+ (\lambda_p - \lambda_p^2)(\|p_1\|^2 + \|p_2\|^2 - \|p_1 - p_2\|^2)$$
$$+ (\lambda_q - \lambda_q^2)(\|q_1\|^2 + \|q_2\|^2 - \|q_1 - q_2\|^2)$$
$$- (1 - \lambda_p - \lambda_q + \lambda_p\lambda_q)(\|p_1\|^2 + \|q_1\|^2 - \|p_1 - q_1\|^2)$$
$$- (\lambda_q - \lambda_p\lambda_q)(\|p_1\|^2 + \|q_2\|^2 - \|p_1 - q_2\|^2)$$
$$- (\lambda_p - \lambda_p\lambda_q)(\|p_2\|^2 + \|q_1\|^2 - \|p_2 - q_1\|^2)$$
$$- \lambda_p\lambda_q(\|p_2\|^2 + \|q_2\|^2 - \|p_2 - q_2\|^2) \tag{VIII}$$

$$= \underbrace{(1 - 1 - 2\lambda_p + \lambda_p + \lambda_p + \lambda_p^2 - \lambda_p^2 + \lambda_q - \lambda_q + \lambda_p\lambda_q - \lambda_p\lambda_q)}_{=0}\|p_1\|^2$$
$$+ \underbrace{(\lambda_p^2 - \lambda_p^2 + \lambda_p - \lambda_p + \lambda_p\lambda_q - \lambda_p\lambda_q)}_{=0}\|p_2\|^2$$

$$+ \underbrace{(1 - 1 - 2\lambda_q + \lambda_q + \lambda_q + \lambda_q^2 - \lambda_q^2 + \lambda_p - \lambda_p + \lambda_p\lambda_q - \lambda_p\lambda_q)}_{=0}\|q_1\|^2$$

$$+ \underbrace{(\lambda_q^2 - \lambda_q^2 + \lambda_q - \lambda_q + \lambda_p\lambda_q - \lambda_p\lambda_q)}_{=0}\|q_2\|^2$$

$$- (\lambda_p - \lambda_p^2)\|p_1 - p_2\|^2 - (\lambda_q - \lambda_q^2)\|q_1 - q_2\|^2 + (1 - \lambda_p - \lambda_q + \lambda_p\lambda_q)\|p_1 - q_1\|^2$$

$$+ (\lambda_q - \lambda_p\lambda_q)\|p_1 - q_2\|^2 + (\lambda_p - \lambda_p\lambda_q)\|p_2 - q_1\|^2 + \lambda_p\lambda_q\|p_2 - q_2\|^2 \qquad \text{(IX)}$$

We obtain Eq. (I) to Eq. (V) using only algebraic manipulations, Eq. (VI) is obtained using the definition of the Euclidean norm and the algebraic definition of the dot product, in Eq. (VII) we use the geometric definition of the dot product and finally in Eq. (VIII) we apply the law of cosines. Eq. (IX) follows by algebraic manipulations. $\square$

*Proof of Lemma 7.* First note that the construction of $f$ succeeds with probability $\rho \in (0, 1]$ by Definition 4. We condition the remaining proof on this event.

From Proposition 6 we now know that

$$\|p - q\|^2 = - (\lambda_p - \lambda_p^2)\|p_1 - p_2\|^2 - (\lambda_q - \lambda_q^2)\|q_1 - q_2\|^2 + (1 - \lambda_p - \lambda_q + \lambda_p\lambda_q)\|p_1 - q_1\|^2$$

$$+ (\lambda_q - \lambda_p\lambda_q)\|p_1 - q_2\|^2 + (\lambda_p - \lambda_p\lambda_q)\|p_2 - q_1\|^2 + \lambda_p\lambda_q\|p_2 - q_2\|^2$$

and

$$\|p' - q'\|^2 = - (\lambda_p - \lambda_p^2)\|f(p_1) - f(p_2)\|^2 - (\lambda_q - \lambda_q^2)\|f(q_1) - f(q_2)\|^2$$

$$+ (1 - \lambda_p - \lambda_q + \lambda_p\lambda_q)\|f(p_1) - f(q_1)\|^2 + (\lambda_q - \lambda_p\lambda_q)\|f(p_1) - f(q_2)\|^2$$

$$+ (\lambda_p - \lambda_p\lambda_q)\|f(p_2) - f(q_1)\|^2 + \lambda_p\lambda_q\|f(p_2) - f(q_2)\|^2.$$

Because every coefficient is non-negative, it can be observed that this sum is maximized under $f$ when

$$\|f(p_1) - f(p_2)\|^2 = (1 - \varepsilon)^2\|p_1 - p_2\|^2,$$
$$\|f(q_1) - f(q_2)\|^2 = (1 - \varepsilon)^2\|q_1 - q_2\|^2,$$
$$\|f(p_1) - f(q_1)\|^2 = (1 + \varepsilon)^2\|p_1 - q_1\|^2,$$
$$\|f(p_1) - f(q_2)\|^2 = (1 + \varepsilon)^2\|p_1 - q_2\|^2,$$
$$\|f(p_2) - f(q_1)\|^2 = (1 + \varepsilon)^2\|p_2 - q_1\|^2$$

and

$$\|f(p_2) - f(q_2)\|^2 = (1 + \varepsilon)^2\|p_2 - q_2\|^2.$$

Using the facts that $(1 + \varepsilon)^2 - (1 - \varepsilon)^2 = 4\varepsilon$, $(\lambda_q - \lambda_q^2) \le \frac{1}{4}$ and $(\lambda_p - \lambda_p^2) \le \frac{1}{4}$, we get that $\|p' - q'\|^2 \le (1 + \varepsilon)^2\|p - q\|^2 + \varepsilon(\|p_1 - p_2\|^2 + \|q_1 - q_2\|^2)$. The lower bound follows analogously. $\square$

*Proof of Theorem 8.* First note that the construction of $f$ and thus also $F$ succeeds with probability $\rho \in (0, 1]$ by Definition 4. We condition the remaining proof on this event.

Let $\tau, \sigma \in T$ be arbitrary polygonal curves and $v_1^\tau, \ldots, v_{|\tau|}^\tau$, respective $v_1^\sigma, \ldots, v_{|\sigma|}^\sigma$ be their vertices, as well as $t_1^\tau, \ldots, t_{|\tau|}^\tau$, respective $t_1^\sigma, \ldots, t_{|\sigma|}^\sigma$ be their instants. Further let

$$g \in \arg\inf_{h \in \mathcal{H}} \max_{t \in [0,1]} \|\tau(t) - \sigma(h(t))\|$$

and

$$g' \in \arg\inf_{h \in \mathcal{H}} \max_{t \in [0,1]} \|F(\tau)(t) - F(\sigma)(h(t))\|.$$

Let $t_1 \in \arg\max_{t \in [0,1]} \|F(\tau)(t) - F(\sigma)(g(t))\|$, there exists an $i \in \{1, \ldots, |\tau|\}$ and a $j \in \{1, \ldots, |\sigma|\}$ with $t_i^\tau \le t_1 \le t_{i+1}^\tau$ and $t_j^\sigma \le g(t_1) \le t_{j+1}^\sigma$, such that we can write

$$F(\tau)(t_1) = \text{lp}\left(\overline{f(v_i^\tau)f(v_{i+1}^\tau)}, \frac{t_1 - t_i^\tau}{t_{i+1}^\tau - t_i^\tau}\right),$$

$$F(\sigma)(g(t_1)) = \mathrm{lp}\left(\overline{f(v_j^\sigma)f(v_{j+1}^\sigma)}, \frac{g(t_1) - t_j^\sigma}{t_{j+1}^\sigma - t_j^\sigma}\right),$$

$$\tau(t_1) = \mathrm{lp}\left(\overline{v_i^\tau v_{i+1}^\tau}, \frac{t_1 - t_i^\tau}{t_{i+1}^\tau - t_i^\tau}\right),$$

and

$$\sigma(g(t_1)) = \mathrm{lp}\left(\overline{v_j^\sigma v_{j+1}^\sigma}, \frac{g(t_1) - t_j^\sigma}{t_{j+1}^\sigma - t_j^\sigma}\right).$$

For each $t_1' \in \arg\max_{t \in [0,1]} \|F(\tau)(t) - F(\sigma)(g'(t))\|$ we obtain:

$$d_F^2(F(\tau), F(\sigma)) = \|F(\tau)(t_1') - F(\sigma)(g'(t_1'))\|^2 \tag{I}$$

$$\leq \|F(\tau)(t_1) - F(\sigma)(g(t_1))\|^2 \tag{II}$$

$$\leq (1+\varepsilon)^2 \|\tau(t_1) - \sigma(g(t_1))\|^2 + \varepsilon\left(\|v_i^\tau - v_{i+1}^\tau\|^2 + \|v_j^\sigma - v_{j+1}^\sigma\|^2\right) \tag{III}$$

$$\leq (1+\varepsilon)^2 \max_{t \in [0,1]} \|\tau(t) - \sigma(g(t))\|^2 + \varepsilon\left(\|v_i^\tau - v_{i+1}^\tau\|^2 + \|v_j^\sigma - v_{j+1}^\sigma\|^2\right)$$

$$\leq (1+\varepsilon)^2 d_F^2(\tau, \sigma) + 2\varepsilon\alpha(\tau, \sigma)^2$$

Eq. (I) follows by definition of $t_1'$ and $g'$, Eq. (II) follows from the fact that $g'$ is an infimum, Eq. (III) follows from an application of Lemma 7 and the last inequality follows from Definition 3 and the definition of $\alpha(\cdot, \cdot)$.

Let $t_2 \in \arg\max_{t \in [0,1]} \|\tau(t) - \sigma(g'(t))\|$, again, there exists an $i \in \{1, \ldots, |\tau|\}$ and a $j \in \{1, \ldots, |\sigma|\}$ with $t_i^\tau \leq t_2 \leq t_{i+1}^\tau$ and $t_j^\sigma \leq g'(t_2) \leq t_{j+1}^\sigma$, such that we can write

$$F(\tau)(t_2) = \mathrm{lp}\left(\overline{f(v_i^\tau)f(v_{i+1}^\tau)}, \frac{t_2 - t_i^\tau}{t_{i+1}^\tau - t_i^\tau}\right),$$

$$F(\sigma)(g'(t_2)) = \mathrm{lp}\left(\overline{f(v_j^\sigma)f(v_{j+1}^\sigma)}, \frac{g'(t_2) - t_j^\sigma}{t_{j+1}^\sigma - t_j^\sigma}\right),$$

$$\tau(t_2) = \mathrm{lp}\left(\overline{v_i^\tau v_{i+1}^\tau}, \frac{t_2 - t_i^\tau}{t_{i+1}^\tau - t_i^\tau}\right),$$

and

$$\sigma(g'(t_2)) = \mathrm{lp}\left(\overline{v_j^\sigma v_{j+1}^\sigma}, \frac{g'(t_2) - t_j^\sigma}{t_{j+1}^\sigma - t_j^\sigma}\right).$$

For each $t_1' \in \arg\max_{t \in [0,1]} \|F(\tau)(t) - F(\sigma)(g'(t))\|$ we obtain:

$$d_F^2(F(\tau), F(\sigma)) = \|F(\tau)(t_1') - F(\sigma)(g'(t_1'))\|^2 \tag{IV}$$

$$\geq \|F(\tau)(t_2) - F(\sigma)(g'(t_2))\|^2 \tag{V}$$

$$\geq (1-\varepsilon)^2 \|\tau(t_2) - \sigma(g'(t_2))\|^2 - \varepsilon\left(\|v_i^\tau - v_{i+1}^\tau\|^2 + \|v_j^\sigma - v_{j+1}^\sigma\|^2\right) \tag{VI}$$

$$\geq (1-\varepsilon)^2 \max_{t \in [0,1]} \|\tau(t) - \sigma(g(t))\|^2 - \varepsilon\left(\|v_i^\tau - v_{i+1}^\tau\|^2 + \|v_j^\sigma - v_{j+1}^\sigma\|^2\right)$$

$$\geq (1-\varepsilon)^2 d_F^2(\tau, \sigma) - 2\varepsilon\alpha(\tau, \sigma)^2$$

Here Eq. (IV) follows by the definition of $g'$ and $t_1'$, Eq. (V) follows, because the term is maximized for $t_1'$, Eq. (VI) follows from an application of Lemma 7 and the last inequality follows from Definition 3 and the definition of $\alpha(\cdot, \cdot)$. $\qquad\square$

*Proof of Theorem 11.* Let $c \in \arg\min_{\tau \in T} \sum_{\tau' \in T} d_F(\tau, \tau')$ be an optimal 1-median for $T$ and let $X(\tau) := d_F(\tau, c)$ be a random variable uniformly distributed over $\tau \in T$. By the uniform distribution and linearity $E[X] = \frac{1}{|T|} \sum_{\tau \in T} d_F(\tau, c)$. Now, let

$$B_{1+\varepsilon} := \left\{\tau \in T \mid d_F(\tau, c) \leq \frac{(1+\varepsilon)}{|T|} \sum_{\tau' \in T} d_F(\tau', c)\right\}.$$

For every $\tau \in B_{1+\varepsilon}$ by the triangle-inequality

$$\sum_{\tau' \in T} d_F(\tau, \tau') \leq \sum_{\tau' \in T} (d_F(\tau', c) + d_F(c, \tau)) \leq (2 + \varepsilon) \sum_{\tau' \in T} d_F(\tau', c).$$

Thus, $\tau$ is at least a $(2 + \varepsilon)$-approximate 1-median for $T$.

For $i \in \{1, \ldots, \ell_s\}$, let $F_i^B$ the event that $s_i \notin B_{1+\varepsilon}$. By Markov's inequality we have that $\Pr[F_i^B] \leq \frac{1}{1+\varepsilon} < 1$.

Further, by independence and choosing $\ell_S \geq \left\lceil \frac{2 \ln(2/\delta)}{\varepsilon} \right\rceil$ the probability that no sample is contained in $B_{1+\varepsilon}$ is bounded by

$$\Pr[F_1^B \wedge \cdots \wedge F_{\ell_S}^B] \leq \frac{1}{(1+\varepsilon)^{\ell_S}} \leq \frac{1}{\exp(\frac{\varepsilon}{2}\ell_S)} \leq \exp(-\varepsilon \ln(2/\delta)/\varepsilon) = \frac{\delta}{2}.$$

Let $c_S \in \arg\min_{\tau \in S} \sum_{\tau' \in T} d_F(\tau, \tau')$. We do not want any *bad* sample $s \in S$ with $\sum_{\tau \in T} d_F(s, \tau) > (1 + \varepsilon) \sum_{\tau \in T} d_F(c_S, \tau)$ to have lower cost with respect to $W$ than $c_S$ . Using Theorem 10 and a union bound over the elements of $S$ and $\ell_W = \frac{64}{\varepsilon^2} \ln(2|S|/\delta)$, the probability for this event is bounded by

$$\sum_{s \in S} \exp\left(-\frac{\varepsilon^2 \ell_W}{64}\right) \leq |S| \exp\left(-\frac{\varepsilon^2 \ell_W}{64}\right) \leq |S| \exp\left(-\ln\left(2|S|/\delta\right)\right) \leq \frac{\delta}{2}.$$

Now, if we take the $s \in S$ that minimizes $\sum_{\tau' \in W} d_F(s, \tau')$, by an application of the union bound, with probability at least $1 - \delta$ it holds that

$$\sum_{\tau' \in T} d_F(s, \tau') \leq (1+\varepsilon) \sum_{\tau' \in T} d_F(c_S, \tau') \leq (1+\varepsilon)(2+\varepsilon) \sum_{\tau' \in T} d_F(c, \tau') \leq (2+4\varepsilon) \sum_{\tau' \in T} d_F(c, \tau').$$

The claim follows by rescaling $\varepsilon$ by $\frac{1}{4}$. $\qquad\square$

*Proof of Theorem 12.* Let $c^* \in \arg\min_{\tau \in T} \sum_{\tau' \in T} d_F(\tau, \tau')$ be an optimal Fréchet 1-median for $T$. For any non-empty set $A$ of curves and a curve $c$ let $\mathrm{cost}(A, c) = \sum_{\tau \in A} d_F(\tau, c)$ denote the cost, i.e., the sum of Fréchet distances to $c$. Let $\Delta = \mathrm{cost}(T, c^*)$ denote the optimal cost. We define a parameter $0 < \gamma(T) := \frac{1}{2} - \nu(T) < \frac{1}{2}$ ($\nu(T)$ will be defined subsequently) which specifies the fraction of outliers as a function of $T$, which may depend on $|T| = n$. We choose the radius $r_1 = \frac{\Delta}{\gamma(T)n}$ which parametrizes the distance of the outliers from the optimal median. Similarly, let $r_2 = 2\varepsilon\frac{\Delta}{n}$. Note that indeed $r_1 > r_2$ as desired, since $\gamma(T), \varepsilon < \frac{1}{2}$. We partition the curves in $T$ according to their contribution relative to the average distance into disjoint sets $T = F \dot\cup M \dot\cup C$ where $F = \{\tau \in T \mid d_F(\tau, c^*) > r_1\}$ are the curves far from $c^*$ , $M = \{\tau \in T \mid r_2 < d_F(\tau, c^*) \leq r_1\}$ are the curves with medium distance, and $C = \{\tau \in T \mid d_F(\tau, c^*) \leq r_2\}$ are the curves that are close to the optimal median.

Note that if $|F| > n \cdot \gamma(T)$ then $\mathrm{cost}(F, c^*) \geq |F| \cdot r_1 > n\gamma(T) \cdot \frac{1}{\gamma(T)} \frac{\Delta}{n} = \Delta$. Together with our assumption this means that we have $(1 - \varepsilon)n\gamma(T) \leq |F| \leq n\gamma(T)$.

Similarly, $\mathrm{cost}(F, c^*) > (1 - \varepsilon)n\gamma(T)\frac{1}{\gamma(T)} \frac{\Delta}{n} = (1 - \varepsilon)\Delta$, which means that the outliers make up a constant fraction of the optimal cost.

Now this implies that $\mathrm{cost}(T \setminus F, c^*) \leq \Delta - (1 - \varepsilon)\Delta = \varepsilon\Delta$, which we can leverage in the following way to bound the number of curves with medium contribution. We have

$$\varepsilon\Delta \geq \mathrm{cost}(T \setminus F, c^*) = \mathrm{cost}(M \dot\cup C, c^*) \geq \mathrm{cost}(M, c^*) \geq |M| \cdot r_2 = |M| \cdot 2\varepsilon\frac{\Delta}{n}.$$

Rearranging yields the desired bound $|M| \leq \frac{\varepsilon\Delta}{2\varepsilon\Delta} \cdot n = \frac{n}{2}$.

Let $A$ be the event that an element sampled uniformly from $T$ is contained in $C$. By the disjoint union, the probability for this event can be bounded by

$$\Pr[A] = \frac{|C|}{n} = \frac{|T| - |M| - |F|}{n} \geq \frac{n - \frac{n}{2} - n\gamma(T)}{n} = \frac{1}{2} - \gamma(T) =: \nu(T).$$

Figure 5: Distributions of curves (for simplicity represented as points) around their median. The red circles represent the radii $r_1, r_2$ defining the sets of far, medium and close curves (cf. proof of Theorem 12, best viewed in color). The left plot shows a "typical" distribution where the median yields a good representative of the data that is robust against outliers. There is a reasonable but not too large number of outliers, that are far away from the center and many curves are close to the optimal median. Such distributions typically arise in physical domains. In such a situation, the sampling algorithm of Theorem 12 yields a $(1+\varepsilon)$-approximation. In the right plot we see a distribution which is much more uniform. Most points are in an annulus about the average distance, there are no far away outliers, and few curves close to the optimal. To find one of the latter, the $(1+\varepsilon)$-approximation needs too many samples. Note however, that the same algorithm yields a $(2+\varepsilon)$-approximation via Theorem 11 that works in general for all inputs.

**Typical distribution with outliers**   **Untypical distribution**

The probability that all of $\ell_S = \frac{1}{1/2 - \gamma(T)} \ln(\frac{2}{\delta})$ i.i.d. uniform samples from $T$ fail to hit $C$ is thus bounded by $(1 - \nu(T))^{\frac{1}{\nu(T)} \ln(\frac{2}{\delta})} \leq e^{-\ln(\frac{2}{\delta})} = \frac{\delta}{2}$.

Thus, with probability at least $1 - \frac{\delta}{2}$ our sample contains at least one $\tilde{c} \in C$ such that $d_F(\tilde{c}, c^*) \leq r_2$. Finally, we have by repeated use of the triangle inequality that

$$\text{cost}(T, \tilde{c}) \leq \text{cost}(T, c^*) + n \cdot d_F(\tilde{c}, c^*) \leq \text{cost}(T, c^*) + n \cdot r_2$$

$$\leq \text{cost}(T, c^*) + n \cdot 2\varepsilon \frac{\text{cost}(T, c^*)}{n} \leq (1 + 2\varepsilon) \text{cost}(T, c^*).$$

As previously we sample a logarithmic number of witnesses $\ell_W = \frac{64}{\varepsilon^2} \ln(\frac{2\ell_S}{\delta})$ such that by Theorem 10 and an application of the union bound the probability that any center that is worse than $\tilde{c}$ by a factor of more than $(1 + \varepsilon)$ has lower cost than $\tilde{c}$ with respect to $W$ is bounded by

$$\sum_{s \in S} \exp\left(-\frac{\varepsilon^2 \ell_W}{64}\right) \leq |S| \cdot \frac{\delta}{2\ell_S} = \frac{\delta}{2}.$$

Thus with probability at least $1 - \delta$ we have that both, our sample $S$ contains a $(1 + 2\varepsilon)$-approximate solution $\tilde{c}$ and any $c' \in S$ that evaluates equal or better than $\tilde{c}$ on the sample $W$ is within $(1 + \varepsilon)$ to the cost of $\tilde{c}$. Thus $\text{cost}(T, c') \leq (1 + \varepsilon)(1 + 2\varepsilon) \text{cost}(T, c^*) \leq (1 + 4\varepsilon) \text{cost}(T, c^*)$.

We conclude the proof by rescaling $\varepsilon$ by $\frac{1}{4}$. $\qquad\square$

*Proof of Theorem 14.* We reduce from the equality test communication problem on bit-strings of size $m$ each. The deterministic communication complexity of this problem is $\Omega(m)$ (Wegener, 2005, Theorem 15.2.2).

In this setting Alice and Bob are given bit-strings $A, B\colon \{1, \ldots, m\} \to \{0, 1\}$ and their task is to decide whether there exists at least one $i \in \{1, \ldots, m\}$ such that $A[i] \neq B[i]$ or not with as little communication as possible. We give a one-way protocol for this problem, where only one message from Alice to Bob is allowed.

In a first step, Alice and Bob construct from their bit-strings polygonal curves $\alpha, \beta$ with $4m$ vertices each. Both curves consist of one gadget per bit. These are either straight-line- or zigzag-gadgets, depending on the value of the respective bit. Specifically, for $i \in \{1, \ldots, m\}$ we define the vertices of $\alpha$:

If $A[i] = 0$ then $v_{4i-3}^{\alpha} := 2i$, $v_{4i-2}^{\alpha} := 2i + 2/3$, $v_{4i-1}^{\alpha} := 2i + 4/3$ and $v_{4i}^{\alpha} := 2i + 2$.
Else, if $A[i] = 1$ then $v_{4i-3}^{\alpha} := 2i$, $v_{4i-2}^{\alpha} := 2i + 2$, $v_{4i-1}^{\alpha} := 2i$ and $v_{4i}^{\alpha} := 2i + 2$.

The vertices $v_{4i-3}^{\beta}, \ldots, v_{4i}^{\beta}$ of $\beta$ are defined analogously.

We claim that

1. $\exists i \in \{1, \ldots, m\} : A[i] \neq B[i] \Rightarrow d_F(\alpha, \beta) \geq 1$ and

2. $\forall i \in \{1, \ldots, m\} : A[i] = B[i] \Rightarrow d_F(\alpha, \beta) = 0$.

To prove the first item, fix an arbitrary $i \in \{1, \ldots, m\}$. W.l.o.g., assume that $A[i] \neq B[i] = 1$. We have the vertices $v_{4i-3}^{\alpha} = 2i$, $v_{4i-2}^{\alpha} = 2i + 2$, $v_{4i-1}^{\alpha} = 2i$ and $v_{4i}^{\alpha} = 2i + 2$, as well as, $v_{4i-3}^{\beta} = 2i$, $v_{4i-2}^{\beta} = 2i + 2/3$, $v_{4i-1}^{\beta} = 2i + 4/3$ and $v_{4i}^{\beta} = 2i + 2$. Let $g \in \arg\inf_{h \in \mathcal{H}} \max_{t \in [0,1]} \|\alpha(t) - \beta(h(t))\|$. Now, assume that $d_F(\alpha, \beta) < 1$. This means, that $g$ must map $v_{4i-3}^{\alpha} = 2i$, $v_{4i-2}^{\alpha} = 2i + 2$ and $v_{4i-1}^{\alpha} = 2i$ to some points that lie closer than $2i + 1 \in \overline{v_{4i-2}^{\beta} v_{4i-1}^{\beta}}$. This is a contradiction, because $g$ is required to be non-decreasing. Thus, in the optimal case $v_{4i-2}^{\alpha}$ and $v_{4i-1}^{\alpha}$ must be mapped to some points infinitesimally close to $2i + 1$.

To prove the second item, observe that by symmetry of the construction, $\alpha$ and $\beta$ represent the same curve and therefore $d_F(\alpha, \beta) = 0$.

Now, suppose there exist oblivious functions $S$ and $E$ not depending on the data such that $d_F(\alpha, \beta) \leq E(S(\alpha), \beta) \leq \eta \cdot d_F(\alpha, \beta)$, for an arbitrary $\eta \in [1, \infty)$.

Alice computes the compressed representation $S(\alpha)$ and communicates $S(\alpha)$ to Bob. Bob evaluates the estimator $E(S(\alpha), \beta)$.

If $E(S(\alpha), \beta) = 0$ then $d_F(\alpha, \beta) \leq E(S(\alpha), \beta) = 0$.

If $E(S(\alpha), \beta) > 0$ then $d_F(\alpha, \beta) \geq E(S(\alpha), \beta)/\eta > 0$.

Thus, Bob can distinguish the above two cases and therefore solve the equality test problem, which implies that $S(\alpha)$ consists of $\Omega(m)$ bits. $\qquad\square$

*Proof of Theorem 15.* We reduce from the set disjointness communication problem on bit strings of size $m$ each. These represent subsets of a common ground set. The randomized communication complexity with public coins is $\Omega(m)$ (Håstad and Wigderson, 2007, Theorem 1.2).

Now, Alice and Bob are given their bit-strings $A, B\colon \{1, \ldots, m\} \to \{0, 1\}$ and their task is to decide whether there exists at least one $i \in \{1, \ldots, m\}$ such that $A[i] = B[i] = 1$ or not with as little communication as possible. We give a one-way protocol for this problem, where only one message from Alice to Bob is allowed.

In a first step, Alice and Bob construct from their bit-strings polygonal curves $\alpha, \beta$ with $4m$ vertices each. Both curves consist of one gadget per bit. These are either straight-line- or notch-gadgets, depending on the value of the respective bit. Thus, for $i \in \{1, \ldots, m\}$ we define the vertices of $\alpha$:

If $A[i] = 0$ then $v_{4i-3}^{\alpha} := (4i, 0)$, $v_{4i-2}^{\alpha} := (4i, 0)$, $v_{4i-1}^{\alpha} := (4i + 4, 0)$ and $v_{4i}^{\alpha} := (4i + 4, 0)$.
Otherwise $v_{4i-3}^{\alpha} := (4i, 0)$, $v_{4i-2}^{\alpha} := (4i, 1)$, $v_{4i-1}^{\alpha} := (4i + 4, 1)$ and $v_{4i}^{\alpha} := (4i + 4, 0)$.

And we define the vertices of $\beta$:

If $B[i] = 0$ then $v_{4i-3}^{\beta} := (4i, 0)$, $v_{4i-2}^{\beta} := (4i, 0)$, $v_{4i-1}^{\beta} := (4i + 4, 0)$ and $v_{4i}^{\beta} := (4i + 4, 0)$.
Otherwise $v_{4i-3}^{\beta} := (4i, 0)$, $v_{4i-2}^{\beta} := (4i, -1)$, $v_{4i-1}^{\beta} := (4i + 4, -1)$ and $v_{4i}^{\beta} := (4i + 4, 0)$.

We claim that

1. $\exists i \in \{1, \ldots, m\} : (A[i] = B[i] = 1) \Rightarrow d_F(\alpha, \beta) \geq 2$ and

2. $\forall i \in \{1, \ldots, m\} : (A[i] = 0 \vee B[i] = 0) \Rightarrow d_F(\alpha, \beta) < \sqrt{2}$.

To prove the first item, fix an arbitrary $i \in \{1, \ldots, m\}$. If $A[i] = B[i] = 1$, we have the vertices $v_{4i-3}^\alpha = (4i, 0), v_{4i-2}^\alpha = (4i, 1), v_{4i-1}^\alpha = (4i+4, 1)$ and $v_{4i}^\alpha = (4i+4, 0)$, as well as, $v_{4i-3}^\beta = (4i, 0), v_{4i-2}^\beta = (4i, -1), v_{4i-1}^\beta = (4i+4, -1)$ and $v_{4i}^\beta = (4i+4, 0)$. Let $g \in \arg\inf_{h \in \mathcal{H}} \max_{t \in [0,1]} \|\alpha(t) - \beta(h(t))\|$. Now, assume that $d_F(\alpha, \beta) < 2$. This means, that $g$ must map $(4i + 2, 1) \in \overline{v_{4i-2}^\alpha v_{4i-1}^\alpha}$ to some point that lies closer than $(4i + 2, -1) \in \overline{v_{4i-2}^\beta v_{4i-1}^\beta}$. This is a contradiction, because the circle of radius 2 around $(4i + 2, 1)$ does only intersect one point of $\beta$, namely $(4i + 2, -1)$. In particular $v_{4i-3}^\beta$ and $v_{4i}^\beta$ have distance $\sqrt{5} > 2$.

To prove the second item, assume w.l.o.g. that $A[i] \neq B[i]$ for all $i \in \{1, \ldots, m\}$. Otherwise $\alpha$ and $\beta$ represent the same curve and have distance 0. Let $m = 1$ and w.l.o.g. assume that $B[1] = 1$. Then we have the vertices $v_1^\alpha = (4, 0), v_2^\alpha = (4, 0), v_3^\alpha = (4 + 4, 0)$ and $v_4^\alpha = (4 + 4, 0)$, as well as $v_1^\beta = (4, 0), v_2^\beta = (4, -1), v_3^\beta = (4 + 4, -1)$ and $v_4^\beta = (4 + 4, 0)$. Let $g$ be a reparameterization that maps $v_1^\alpha$ to $v_1^\beta$ and $v_4^\alpha$ to $v_4^\beta$, as well as $\overline{v_1^\beta v_2^\beta}$ and $\overline{v_3^\beta v_4^\beta}$ to some infinitesimally small sub-segment of $\overline{v_1^\alpha v_4^\alpha}$ each. Since these sub-segments have length less than 1 each, any point of these is mapped to a point within distance less than $\sqrt{2}$. Now, let $g$ map the remaining segment $\overline{v_2^\beta v_3^\beta}$ of $\beta$ linearly to the remaining middle sub-segment of $\overline{v_1^\alpha v_4^\alpha}$ of $\alpha$. Since this remaining sub-segment has length larger than 2, again any point is mapped to a point within distance less than $\sqrt{2}$. Since we can inductively apply this argument for any $m > 1$, i.e., any number of gadgets, we conclude that $d_F(\alpha, \beta) < \sqrt{2}$.

Now, suppose there exist oblivious randomized functions $S$ and $E$ not depending on the data such that $d_F(\alpha, \beta) \leq E(S(\alpha), \beta) \leq \eta \cdot d_F(\alpha, \beta)$ with constant probability, for an arbitrary $\eta \in [1, \sqrt{2}]$.

Alice computes the compressed representation $S(\alpha)$ using some of the public coins and communicates $S(\alpha)$ to Bob. Bob evaluates the estimator $E(S(\alpha), \beta)$.

If $E(S(\alpha), \beta) < 2$ then with constant probability $d_F(\alpha, \beta) \leq E(S(\alpha), \beta) < 2$.

If $E(S(\alpha), \beta) \geq 2$ then with constant probability $d_F(\alpha, \beta) \geq E(S(\alpha), \beta)/\sqrt{2} \geq \sqrt{2}$.

Thus, Bob can distinguish the above two cases and therefore solve the set disjointness problem with constant probability, which implies that $S(\alpha)$ consists of $\Omega(m)$ bits. $\qquad \square$

[Supplementary Material 2]

## A Omitted material

*Proof of Proposition 6.* We have:

$$\|p - q\|^2 = \sum_{i=1}^{d}[(p_{1,i} - \lambda_p(p_{1,i} - p_{2,i})) - (q_{1,i} - \lambda_q(q_{1,i} - q_{2,i}))]^2 \tag{I}$$

$$= \sum_{i=1}^{d}[(p_{1,i} - \lambda_p(p_{1,i} - p_{2,i}))^2 - 2(p_{1,i} - \lambda_p(p_{1,i} - p_{2,i}))(q_{1,i} - \lambda_q(q_{1,i} - q_{2,i}))$$
$$+ (q_{1,i} - \lambda_q(q_{1,i} - q_{2,i}))^2] \tag{II}$$

$$= \sum_{i=1}^{d}[p_{1,i}^2 - 2\lambda_p p_{1,i}(p_{1,i} - p_{2,i}) + \lambda_p^2(p_{1,i} - p_{2,i})^2$$
$$- 2(p_{1,i} - \lambda_p(p_{1,i} - p_{2,i}))(q_{1,i} - \lambda_q(q_{1,i} - q_{2,i})) + q_{1,i}^2 - 2\lambda_q q_{1,i}(q_{1,i} - q_{2,i})$$
$$+ \lambda_q^2(q_{1,i} - q_{2,i})^2] \tag{III}$$

$$= \sum_{i=1}^{d}[p_{1,i}^2 - 2\lambda_p p_{1,i}^2 + 2\lambda_p p_{1,i}p_{2,i} + \lambda_p^2 p_{1,i}^2 - 2\lambda_p^2 p_{1,i}p_{2,i} + \lambda_p^2 p_{2,i}^2 - 2p_{1,i}q_{1,i}$$
$$+ 2\lambda_q p_{1,i}(q_{1,i} - q_{2,i}) + 2\lambda_p q_{1,i}(p_{1,i} - p_{2,i}) - 2\lambda_p\lambda_q(p_{1,i} - p_{2,i})(q_{1,i} - q_{2,i})$$
$$+ q_{1,i}^2 - 2\lambda_q q_{1,i}^2 + 2\lambda_q q_{1,i}q_{2,i} + \lambda_q^2 q_{1,i}^2 - 2\lambda_q^2 q_{1,i}q_{2,i} + \lambda_q^2 q_{2,i}^2] \tag{IV}$$

$$= \sum_{i=1}^{d}[(1 - 2\lambda_p + \lambda_p^2)p_{1,i}^2 + \lambda_p^2 p_{2,i}^2 + (1 - 2\lambda_q + \lambda_q^2)q_{1,i}^2 + \lambda_q^2 q_{2,i}^2$$
$$+ 2\lambda_p(1 - \lambda_p)p_{1,i}p_{2,i} - 2p_{1,i}q_{1,i} + 2\lambda_q p_{1,i}q_{1,i} - 2\lambda_p p_{1,i}q_{2,i} + 2\lambda_p p_{1,i}q_{1,i}$$
$$- 2\lambda_p p_{2,i}q_{1,i} - 2\lambda_p\lambda_q p_{1,i}q_{1,i} + 2\lambda_p\lambda_q p_{2,i}q_{1,i} + 2\lambda_p\lambda_q p_{1,i}q_{2,i}$$
$$- 2\lambda_p\lambda_q p_{2,i}q_{2,i} + 2\lambda_q(1 - \lambda_q)q_{1,i}q_{2,i}] \tag{V}$$

$$= (1 - \lambda_p)^2\|p_1\|^2 + \lambda_p^2\|p_2\|^2 + (1 - \lambda_q)^2\|q_1\|^2 + \lambda_q^2\|q_2\|^2$$
$$+ 2\lambda_p(1 - \lambda_p)\langle p_1, p_2\rangle + 2(\lambda_p + \lambda_q - \lambda_p\lambda_q - 1)\langle p_1, q_1\rangle$$
$$+ 2(\lambda_p\lambda_q - \lambda_q)\langle p_1, q_2\rangle + 2(\lambda_p\lambda_q - \lambda_p)\langle p_2, q_1\rangle - 2\lambda_p\lambda_q\langle p_2, q_2\rangle$$
$$+ 2\lambda_q(1 - \lambda_q)\langle q_1, q_2\rangle \tag{VI}$$

$$= (1 - \lambda_p)^2\|p_1\|^2 + \lambda_p^2\|p_2\|^2 + (1 - \lambda_q)^2\|q_1\|^2 + \lambda_q^2\|q_2\|^2$$
$$+ 2\lambda_p(1 - \lambda_p)\|p_1\|\|p_2\|\cos\sphericalangle(p_1, p_2)$$
$$+ 2(\lambda_p + \lambda_q - \lambda_p\lambda_q - 1)\|p_1\|\|q_1\|\cos\sphericalangle(p_1, q_1)$$
$$+ 2(\lambda_p\lambda_q - \lambda_q)\|p_1\|\|q_2\|\cos\sphericalangle(p_1, q_2)$$
$$+ 2(\lambda_p\lambda_q - \lambda_p)\|p_2\|\|q_1\|\cos\sphericalangle(p_2, q_1)$$
$$- 2\lambda_p\lambda_q\|p_2\|\|q_2\|\cos\sphericalangle(p_2, q_2) + 2\lambda_q(1 - \lambda_q)\|q_1\|\|q_2\|\cos\sphericalangle(q_1, q_2) \tag{VII}$$

$$= (1 - \lambda_p)^2\|p_1\|^2 + \lambda_p^2\|p_2\|^2 + (1 - \lambda_q)^2\|q_1\|^2 + \lambda_q^2\|q_2\|^2$$
$$+ (\lambda_p - \lambda_p^2)(\|p_1\|^2 + \|p_2\|^2 - \|p_1 - p_2\|^2)$$
$$+ (\lambda_q - \lambda_q^2)(\|q_1\|^2 + \|q_2\|^2 - \|q_1 - q_2\|^2)$$
$$- (1 - \lambda_p - \lambda_q + \lambda_p\lambda_q)(\|p_1\|^2 + \|q_1\|^2 - \|p_1 - q_1\|^2)$$
$$- (\lambda_q - \lambda_p\lambda_q)(\|p_1\|^2 + \|q_2\|^2 - \|p_1 - q_2\|^2)$$
$$- (\lambda_p - \lambda_p\lambda_q)(\|p_2\|^2 + \|q_1\|^2 - \|p_2 - q_1\|^2)$$
$$- \lambda_p\lambda_q(\|p_2\|^2 + \|q_2\|^2 - \|p_2 - q_2\|^2) \tag{VIII}$$

$$= \underbrace{(1 - 1 - 2\lambda_p + \lambda_p + \lambda_p + \lambda_p^2 - \lambda_p^2 + \lambda_q - \lambda_q + \lambda_p\lambda_q - \lambda_p\lambda_q)}_{=0}\|p_1\|^2$$

$$+ \underbrace{(\lambda_p^2 - \lambda_p^2 + \lambda_p - \lambda_p + \lambda_p\lambda_q - \lambda_p\lambda_q)}_{=0}\|p_2\|^2$$

$$+ \underbrace{(1 - 1 - 2\lambda_q + \lambda_q + \lambda_q + \lambda_q^2 - \lambda_q^2 + \lambda_p - \lambda_p + \lambda_p\lambda_q - \lambda_p\lambda_q)}_{=0}\|q_1\|^2$$

$$+ \underbrace{(\lambda_q^2 - \lambda_q^2 + \lambda_q - \lambda_q + \lambda_p\lambda_q - \lambda_p\lambda_q)}_{=0}\|q_2\|^2$$

$$- (\lambda_p - \lambda_p^2)\|p_1 - p_2\|^2 - (\lambda_q - \lambda_q^2)\|q_1 - q_2\|^2 + (1 - \lambda_p - \lambda_q + \lambda_p\lambda_q)\|p_1 - q_1\|^2$$

$$+ (\lambda_q - \lambda_p\lambda_q)\|p_1 - q_2\|^2 + (\lambda_p - \lambda_p\lambda_q)\|p_2 - q_1\|^2 + \lambda_p\lambda_q\|p_2 - q_2\|^2 \qquad \text{(IX)}$$

We obtain Eq. (I) to Eq. (V) using only algebraic manipulations, Eq. (VI) is obtained using the definition of the Euclidean norm and the algebraic definition of the dot product, in Eq. (VII) we use the geometric definition of the dot product and finally in Eq. (VIII) we apply the law of cosines. Eq. (IX) follows by algebraic manipulations. □

*Proof of Lemma 7.* First note that the construction of $f$ succeeds with probability $\rho \in (0, 1]$ by Definition 4. We condition the remaining proof on this event.

From Proposition 6 we now know that

$$\|p - q\|^2 = -(\lambda_p - \lambda_p^2)\|p_1 - p_2\|^2 - (\lambda_q - \lambda_q^2)\|q_1 - q_2\|^2 + (1 - \lambda_p - \lambda_q + \lambda_p\lambda_q)\|p_1 - q_1\|^2$$
$$+ (\lambda_q - \lambda_p\lambda_q)\|p_1 - q_2\|^2 + (\lambda_p - \lambda_p\lambda_q)\|p_2 - q_1\|^2 + \lambda_p\lambda_q\|p_2 - q_2\|^2$$

and

$$\|p' - q'\|^2 = -(\lambda_p - \lambda_p^2)\|f(p_1) - f(p_2)\|^2 - (\lambda_q - \lambda_q^2)\|f(q_1) - f(q_2)\|^2$$
$$+ (1 - \lambda_p - \lambda_q + \lambda_p\lambda_q)\|f(p_1) - f(q_1)\|^2 + (\lambda_q - \lambda_p\lambda_q)\|f(p_1) - f(q_2)\|^2$$
$$+ (\lambda_p - \lambda_p\lambda_q)\|f(p_2) - f(q_1)\|^2 + \lambda_p\lambda_q\|f(p_2) - f(q_2)\|^2.$$

Because every coefficient is non-negative, it can be observed that this sum is maximized under $f$ when

$$\|f(p_1) - f(p_2)\|^2 = (1 - \varepsilon)^2\|p_1 - p_2\|^2,$$
$$\|f(q_1) - f(q_2)\|^2 = (1 - \varepsilon)^2\|q_1 - q_2\|^2,$$
$$\|f(p_1) - f(q_1)\|^2 = (1 + \varepsilon)^2\|p_1 - q_1\|^2,$$
$$\|f(p_1) - f(q_2)\|^2 = (1 + \varepsilon)^2\|p_1 - q_2\|^2,$$
$$\|f(p_2) - f(q_1)\|^2 = (1 + \varepsilon)^2\|p_2 - q_1\|^2$$

and

$$\|f(p_2) - f(q_2)\|^2 = (1 + \varepsilon)^2\|p_2 - q_2\|^2.$$

Using the facts that $(1 + \varepsilon)^2 - (1 - \varepsilon)^2 = 4\varepsilon$, $(\lambda_q - \lambda_q^2) \leq \frac{1}{4}$ and $(\lambda_p - \lambda_p^2) \leq \frac{1}{4}$, we get that $\|p' - q'\|^2 \leq (1 + \varepsilon)^2\|p - q\|^2 + \varepsilon(\|p_1 - p_2\|^2 + \|q_1 - q_2\|^2)$. The lower bound follows analogously. □

*Proof of Theorem 8.* First note that the construction of $f$ and thus also $F$ succeeds with probability $\rho \in (0, 1]$ by Definition 4. We condition the remaining proof on this event.

Let $\tau, \sigma \in T$ be arbitrary polygonal curves and $v_1^\tau, \dots, v_{|\tau|}^\tau$, respective $v_1^\sigma, \dots, v_{|\sigma|}^\sigma$ be their vertices, as well as $t_1^\tau, \dots, t_{|\tau|}^\tau$, respective $t_1^\sigma, \dots, t_{|\sigma|}^\sigma$ be their instants. Further let

$$g \in \arg\inf_{h \in \mathcal{H}} \max_{t \in [0,1]} \|\tau(t) - \sigma(h(t))\|$$

and

$$g' \in \arg\inf_{h \in \mathcal{H}} \max_{t \in [0,1]} \|F(\tau)(t) - F(\sigma)(h(t))\|.$$

Let $t_1 \in \arg\max_{t \in [0,1]} \|F(\tau)(t) - F(\sigma)(g(t))\|$, there exists an $i \in \{1, \dots, |\tau|\}$ and a $j \in \{1, \dots, |\sigma|\}$ with $t_i^\tau \leq t_1 \leq t_{i+1}^\tau$ and $t_j^\sigma \leq g(t_1) \leq t_{j+1}^\sigma$, such that we can write

$$F(\tau)(t_1) = \text{lp}\left(\overline{f(v_i^\tau)f(v_{i+1}^\tau)}, \frac{t_1 - t_i^\tau}{t_{i+1}^\tau - t_i^\tau}\right),$$

$$F(\sigma)(g(t_1)) = \mathrm{lp}\left(\overline{f(v_j^\sigma)f(v_{j+1}^\sigma)}, \frac{g(t_1) - t_j^\sigma}{t_{j+1}^\sigma - t_j^\sigma}\right),$$

$$\tau(t_1) = \mathrm{lp}\left(\overline{v_i^\tau v_{i+1}^\tau}, \frac{t_1 - t_i^\tau}{t_{i+1}^\tau - t_i^\tau}\right),$$

and

$$\sigma(g(t_1)) = \mathrm{lp}\left(\overline{v_j^\sigma v_{j+1}^\sigma}, \frac{g(t_1) - t_j^\sigma}{t_{j+1}^\sigma - t_j^\sigma}\right).$$

For each $t_1' \in \arg\max_{t\in[0,1]} \|F(\tau)(t) - F(\sigma)(g'(t))\|$ we obtain:

$$d_F^2(F(\tau), F(\sigma)) = \|F(\tau)(t_1') - F(\sigma)(g'(t_1'))\|^2 \tag{I}$$
$$\leq \|F(\tau)(t_1) - F(\sigma)(g(t_1))\|^2 \tag{II}$$
$$\leq (1+\varepsilon)^2 \|\tau(t_1) - \sigma(g(t_1))\|^2 + \varepsilon\left(\|v_i^\tau - v_{i+1}^\tau\|^2 + \|v_j^\sigma - v_{j+1}^\sigma\|^2\right) \tag{III}$$
$$\leq (1+\varepsilon)^2 \max_{t\in[0,1]} \|\tau(t) - \sigma(g(t))\|^2 + \varepsilon\left(\|v_i^\tau - v_{i+1}^\tau\|^2 + \|v_j^\sigma - v_{j+1}^\sigma\|^2\right)$$
$$\leq (1+\varepsilon)^2 d_F^2(\tau, \sigma) + 2\varepsilon\alpha(\tau, \sigma)^2$$

Eq. (I) follows by definition of $t_1'$ and $g'$, Eq. (II) follows from the fact that $g'$ is an infimum, Eq. (III) follows from an application of Lemma 7 and the last inequality follows from Definition 3 and the definition of $\alpha(\cdot, \cdot)$.

Let $t_2 \in \arg\max_{t\in[0,1]} \|\tau(t) - \sigma(g'(t))\|$, again, there exists an $i \in \{1, \ldots, |\tau|\}$ and a $j \in \{1, \ldots, |\sigma|\}$ with $t_i^\tau \leq t_2 \leq t_{i+1}^\tau$ and $t_j^\sigma \leq g'(t_2) \leq t_{j+1}^\sigma$, such that we can write

$$F(\tau)(t_2) = \mathrm{lp}\left(\overline{f(v_i^\tau)f(v_{i+1}^\tau)}, \frac{t_2 - t_i^\tau}{t_{i+1}^\tau - t_i^\tau}\right),$$

$$F(\sigma)(g'(t_2)) = \mathrm{lp}\left(\overline{f(v_j^\sigma)f(v_{j+1}^\sigma)}, \frac{g'(t_2) - t_j^\sigma}{t_{j+1}^\sigma - t_j^\sigma}\right),$$

$$\tau(t_2) = \mathrm{lp}\left(\overline{v_i^\tau v_{i+1}^\tau}, \frac{t_2 - t_i^\tau}{t_{i+1}^\tau - t_i^\tau}\right),$$

and

$$\sigma(g'(t_2)) = \mathrm{lp}\left(\overline{v_j^\sigma v_{j+1}^\sigma}, \frac{g'(t_2) - t_j^\sigma}{t_{j+1}^\sigma - t_j^\sigma}\right).$$

For each $t_1' \in \arg\max_{t\in[0,1]} \|F(\tau)(t) - F(\sigma)(g'(t))\|$ we obtain:

$$d_F^2(F(\tau), F(\sigma)) = \|F(\tau)(t_1') - F(\sigma)(g'(t_1'))\|^2 \tag{IV}$$
$$\geq \|F(\tau)(t_2) - F(\sigma)(g'(t_2))\|^2 \tag{V}$$
$$\geq (1-\varepsilon)^2 \|\tau(t_2) - \sigma(g'(t_2))\|^2 - \varepsilon\left(\|v_i^\tau - v_{i+1}^\tau\|^2 + \|v_j^\sigma - v_{j+1}^\sigma\|^2\right) \tag{VI}$$
$$\geq (1-\varepsilon)^2 \max_{t\in[0,1]} \|\tau(t) - \sigma(g(t))\|^2 - \varepsilon\left(\|v_i^\tau - v_{i+1}^\tau\|^2 + \|v_j^\sigma - v_{j+1}^\sigma\|^2\right)$$
$$\geq (1-\varepsilon)^2 d_F^2(\tau, \sigma) - 2\varepsilon\alpha(\tau, \sigma)^2$$

Here Eq. (IV) follows by the definition of $g'$ and $t_1'$, Eq. (V) follows, because the term is maximized for $t_1'$, Eq. (VI) follows from an application of Lemma 7 and the last inequality follows from Definition 3 and the definition of $\alpha(\cdot, \cdot)$. $\square$

*Proof of Theorem 11.* Let $c \in \arg\min_{\tau\in T} \sum_{\tau'\in T} d_F(\tau, \tau')$ be an optimal 1-median for $T$ and let $X(\tau) := d_F(\tau, c)$ be a random variable uniformly distributed over $\tau \in T$. By the uniform distribution and linearity $E[X] = \frac{1}{|T|} \sum_{\tau\in T} d_F(\tau, c)$. Now, let

$$B_{1+\varepsilon} := \left\{\tau \in T \mid d_F(\tau, c) \leq \frac{(1+\varepsilon)}{|T|} \sum_{\tau'\in T} d_F(\tau', c)\right\}.$$

For every $\tau \in B_{1+\varepsilon}$ by the triangle-inequality

$$\sum_{\tau' \in T} d_F(\tau, \tau') \leq \sum_{\tau' \in T} (d_F(\tau', c) + d_F(c, \tau)) \leq (2 + \varepsilon) \sum_{\tau' \in T} d_F(\tau', c).$$

Thus, $\tau$ is at least a $(2 + \varepsilon)$-approximate 1-median for $T$.

For $i \in \{1, \ldots, \ell_s\}$, let $F_i^B$ the event that $s_i \notin B_{1+\varepsilon}$. By Markov's inequality we have that $\Pr[F_i^B] \leq \frac{1}{1+\varepsilon} < 1$.

Further, by independence and choosing $\ell_S \geq \left\lceil \frac{2 \ln(2/\delta)}{\varepsilon} \right\rceil$ the probability that no sample is contained in $B_{1+\varepsilon}$ is bounded by

$$\Pr[F_1^B \wedge \cdots \wedge F_{\ell_S}^B] \leq \frac{1}{(1+\varepsilon)^{\ell_S}} \leq \frac{1}{\exp(\frac{\varepsilon}{2}\ell_S)} \leq \exp(-\varepsilon \ln(2/\delta)/\varepsilon) = \frac{\delta}{2}.$$

Let $c_S \in \arg\min_{\tau \in S} \sum_{\tau' \in T} d_F(\tau, \tau')$. We do not want any *bad* sample $s \in S$ with $\sum_{\tau \in T} d_F(s, \tau) > (1 + \varepsilon) \sum_{\tau \in T} d_F(c_S, \tau)$ to have lower cost with respect to $W$ than $c_S$. Using Theorem 10 and a union bound over the elements of $S$ and $\ell_W = \frac{64}{\varepsilon^2} \ln(2|S|/\delta)$, the probability for this event is bounded by

$$\sum_{s \in S} \exp\left(-\frac{\varepsilon^2 \ell_W}{64}\right) \leq |S| \exp\left(-\frac{\varepsilon^2 \ell_W}{64}\right) \leq |S| \exp\left(-\ln(2|S|/\delta)\right) \leq \frac{\delta}{2}.$$

Now, if we take the $s \in S$ that minimizes $\sum_{\tau' \in W} d_F(s, \tau')$, by an application of the union bound, with probability at least $1 - \delta$ it holds that

$$\sum_{\tau' \in T} d_F(s, \tau') \leq (1+\varepsilon) \sum_{\tau' \in T} d_F(c_S, \tau') \leq (1+\varepsilon)(2+\varepsilon) \sum_{\tau' \in T} d_F(c, \tau') \leq (2+4\varepsilon) \sum_{\tau' \in T} d_F(c, \tau').$$

The claim follows by rescaling $\varepsilon$ by $\frac{1}{4}$. $\qquad\square$

*Proof of Theorem 12.* Let $c^* \in \arg\min_{\tau \in T} \sum_{\tau' \in T} d_F(\tau, \tau')$ be an optimal Fréchet 1-median for $T$. For any non-empty set $A$ of curves and a curve $c$ let $\text{cost}(A, c) = \sum_{\tau \in A} d_F(\tau, c)$ denote the cost, i.e., the sum of Fréchet distances to $c$. Let $\Delta = \text{cost}(T, c^*)$ denote the optimal cost. We define a parameter $0 < \gamma(T) := \frac{1}{2} - \nu(T) < \frac{1}{2}$ ($\nu(T)$ will be defined subsequently) which specifies the fraction of outliers as a function of $T$, which may depend on $|T| = n$. We choose the radius $r_1 = \frac{\Delta}{\gamma(T)n}$ which parametrizes the distance of the outliers from the optimal median. Similarly, let $r_2 = 2\varepsilon \frac{\Delta}{n}$. Note that indeed $r_1 > r_2$ as desired, since $\gamma(T), \varepsilon < \frac{1}{2}$. We partition the curves in $T$ according to their contribution relative to the average distance into disjoint sets $T = F \dot\cup M \dot\cup C$ where $F = \{\tau \in T \mid d_F(\tau, c^*) > r_1\}$ are the curves far from $c^*$, $M = \{\tau \in T \mid r_2 < d_F(\tau, c^*) \leq r_1\}$ are the curves with medium distance, and $C = \{\tau \in T \mid d_F(\tau, c^*) \leq r_2\}$ are the curves that are close to the optimal median.

Note that if $|F| > n \cdot \gamma(T)$ then $\text{cost}(F, c^*) \geq |F| \cdot r_1 > n\gamma(T) \cdot \frac{1}{\gamma(T)} \frac{\Delta}{n} = \Delta$. Together with our assumption this means that we have $(1 - \varepsilon)n\gamma(T) \leq |F| \leq n\gamma(T)$.

Similarly, $\text{cost}(F, c^*) > (1 - \varepsilon)n\gamma(T)\frac{1}{\gamma(T)}\frac{\Delta}{n} = (1 - \varepsilon)\Delta$, which means that the outliers make up a constant fraction of the optimal cost.

Now this implies that $\text{cost}(T \setminus F, c^*) \leq \Delta - (1 - \varepsilon)\Delta = \varepsilon\Delta$, which we can leverage in the following way to bound the number of curves with medium contribution. We have

$$\varepsilon\Delta \geq \text{cost}(T \setminus F, c^*) = \text{cost}(M \dot\cup C, c^*) \geq \text{cost}(M, c^*) \geq |M| \cdot r_2 = |M| \cdot 2\varepsilon\frac{\Delta}{n}.$$

Rearranging yields the desired bound $|M| \leq \frac{\varepsilon\Delta}{2\varepsilon\Delta} \cdot n = \frac{n}{2}$.

Let $A$ be the event that an element sampled uniformly from $T$ is contained in $C$. By the disjoint union, the probability for this event can be bounded by

$$\Pr[A] = \frac{|C|}{n} = \frac{|T| - |M| - |F|}{n} \geq \frac{n - \frac{n}{2} - n\gamma(T)}{n} = \frac{1}{2} - \gamma(T) =: \nu(T).$$

Figure 5: Distributions of curves (for simplicity represented as points) around their median. The red circles represent the radii $r_1, r_2$ defining the sets of far, medium and close curves (cf. proof of Theorem 12, best viewed in color). The left plot shows a "typical" distribution where the median yields a good representative of the data that is robust against outliers. There is a reasonable but not too large number of outliers, that are far away from the center and many curves are close to the optimal median. Such distributions typically arise in physical domains. In such a situation, the sampling algorithm of Theorem 12 yields a $(1 + \varepsilon)$-approximation. In the right plot we see a distribution which is much more uniform. Most points are in an annulus about the average distance, there are no far away outliers, and few curves close to the optimal. To find one of the latter, the $(1 + \varepsilon)$-approximation needs too many samples. Note however, that the same algorithm yields a $(2 + \varepsilon)$-approximation via Theorem 11 that works in general for all inputs.

**Typical distribution with outliers**          **Untypical distribution**

The probability that all of $\ell_S = \frac{1}{1/2 - \gamma(T)} \ln(\frac{2}{\delta})$ i.i.d. uniform samples from $T$ fail to hit $C$ is thus bounded by $(1 - \nu(T))^{\frac{1}{\nu(T)} \ln(\frac{2}{\delta})} \leq e^{-\ln(\frac{2}{\delta})} = \frac{\delta}{2}$.

Thus, with probability at least $1 - \frac{\delta}{2}$ our sample contains at least one $\tilde{c} \in C$ such that $d_F(\tilde{c}, c^*) \leq r_2$. Finally, we have by repeated use of the triangle inequality that

$$\text{cost}(T, \tilde{c}) \leq \text{cost}(T, c^*) + n \cdot d_F(\tilde{c}, c^*) \leq \text{cost}(T, c^*) + n \cdot r_2$$

$$\leq \text{cost}(T, c^*) + n \cdot 2\varepsilon \frac{\text{cost}(T, c^*)}{n} \leq (1 + 2\varepsilon) \text{cost}(T, c^*).$$

As previously we sample a logarithmic number of witnesses $\ell_W = \frac{64}{\varepsilon^2} \ln(\frac{2\ell_S}{\delta})$ such that by Theorem 10 and an application of the union bound the probability that any center that is worse than $\tilde{c}$ by a factor of more than $(1 + \varepsilon)$ has lower cost than $\tilde{c}$ with respect to $W$ is bounded by

$$\sum_{s \in S} \exp\left(-\frac{\varepsilon^2 \ell_W}{64}\right) \leq |S| \cdot \frac{\delta}{2\ell_S} = \frac{\delta}{2}.$$

Thus with probability at least $1 - \delta$ we have that both, our sample $S$ contains a $(1 + 2\varepsilon)$-approximate solution $\tilde{c}$ and any $c' \in S$ that evaluates equal or better than $\tilde{c}$ on the sample $W$ is within $(1 + \varepsilon)$ to the cost of $\tilde{c}$. Thus $\text{cost}(T, c') \leq (1 + \varepsilon)(1 + 2\varepsilon) \text{cost}(T, c^*) \leq (1 + 4\varepsilon) \text{cost}(T, c^*)$.

We conclude the proof by rescaling $\varepsilon$ by $\frac{1}{4}$. $\qquad\qquad\qquad\qquad\qquad\qquad\qquad\qquad\qquad\qquad\square$

*Proof of Theorem 14.* We reduce from the equality test communication problem on bit-strings of size $m$ each. The deterministic communication complexity of this problem is $\Omega(m)$ (Wegener, 2005, Theorem 15.2.2).

In this setting Alice and Bob are given bit-strings $A, B \colon \{1, \ldots, m\} \to \{0, 1\}$ and their task is to decide whether there exists at least one $i \in \{1, \ldots, m\}$ such that $A[i] \neq B[i]$ or not with as little communication as possible. We give a one-way protocol for this problem, where only one message from Alice to Bob is allowed.

In a first step, Alice and Bob construct from their bit-strings polygonal curves $\alpha, \beta$ with $4m$ vertices each. Both curves consist of one gadget per bit. These are either straight-line- or zigzag-gadgets, depending on the value of the respective bit. Specifically, for $i \in \{1, \ldots, m\}$ we define the vertices of $\alpha$:

If $A[i] = 0$ then $v_{4i-3}^{\alpha} := 2i$, $v_{4i-2}^{\alpha} := 2i + 2/3$, $v_{4i-1}^{\alpha} := 2i + 4/3$ and $v_{4i}^{\alpha} := 2i + 2$.
Else, if $A[i] = 1$ then $v_{4i-3}^{\alpha} := 2i$, $v_{4i-2}^{\alpha} := 2i + 2$, $v_{4i-1}^{\alpha} := 2i$ and $v_{4i}^{\alpha} := 2i + 2$.

The vertices $v_{4i-3}^{\beta}, \ldots, v_{4i}^{\beta}$ of $\beta$ are defined analogously.

We claim that

1. $\exists i \in \{1, \ldots, m\} : A[i] \neq B[i] \Rightarrow d_F(\alpha, \beta) \geq 1$ and

2. $\forall i \in \{1, \ldots, m\} : A[i] = B[i] \Rightarrow d_F(\alpha, \beta) = 0$.

To prove the first item, fix an arbitrary $i \in \{1, \ldots, m\}$. W.l.o.g., assume that $A[i] \neq B[i] = 1$. We have the vertices $v_{4i-3}^{\alpha} = 2i$, $v_{4i-2}^{\alpha} = 2i + 2$, $v_{4i-1}^{\alpha} = 2i$ and $v_{4i}^{\alpha} = 2i + 2$, as well as, $v_{4i-3}^{\beta} = 2i$, $v_{4i-2}^{\beta} = 2i + 2/3$, $v_{4i-1}^{\beta} = 2i + 4/3$ and $v_{4i}^{\beta} = 2i + 2$. Let $g \in \arg\inf_{h \in \mathcal{H}} \max_{t \in [0,1]} \|\alpha(t) - \beta(h(t))\|$. Now, assume that $d_F(\alpha, \beta) < 1$. This means, that $g$ must map $v_{4i-3}^{\alpha} = 2i$, $v_{4i-2}^{\alpha} = 2i + 2$ and $v_{4i-1}^{\alpha} = 2i$ to some points that lie closer than $2i + 1 \in \overline{v_{4i-2}^{\beta} v_{4i-1}^{\beta}}$. This is a contradiction, because $g$ is required to be non-decreasing. Thus, in the optimal case $v_{4i-2}^{\alpha}$ and $v_{4i-1}^{\alpha}$ must be mapped to some points infinitesimally close to $2i + 1$.

To prove the second item, observe that by symmetry of the construction, $\alpha$ and $\beta$ represent the same curve and therefore $d_F(\alpha, \beta) = 0$.

Now, suppose there exist oblivious functions $S$ and $E$ not depending on the data such that $d_F(\alpha, \beta) \leq E(S(\alpha), \beta) \leq \eta \cdot d_F(\alpha, \beta)$, for an arbitrary $\eta \in [1, \infty)$.

Alice computes the compressed representation $S(\alpha)$ and communicates $S(\alpha)$ to Bob. Bob evaluates the estimator $E(S(\alpha), \beta)$.

If $E(S(\alpha), \beta) = 0$ then $d_F(\alpha, \beta) \leq E(S(\alpha), \beta) = 0$.

If $E(S(\alpha), \beta) > 0$ then $d_F(\alpha, \beta) \geq E(S(\alpha), \beta)/\eta > 0$.

Thus, Bob can distinguish the above two cases and therefore solve the equality test problem, which implies that $S(\alpha)$ consists of $\Omega(m)$ bits. $\qquad \square$

*Proof of Theorem 15.* We reduce from the set disjointness communication problem on bit strings of size $m$ each. These represent subsets of a common ground set. The randomized communication complexity with public coins is $\Omega(m)$ (Håstad and Wigderson, 2007, Theorem 1.2).

Now, Alice and Bob are given their bit-strings $A, B \colon \{1, \ldots, m\} \to \{0, 1\}$ and their task is to decide whether there exists at least one $i \in \{1, \ldots, m\}$ such that $A[i] = B[i] = 1$ or not with as little communication as possible. We give a one-way protocol for this problem, where only one message from Alice to Bob is allowed.

In a first step, Alice and Bob construct from their bit-strings polygonal curves $\alpha, \beta$ with $4m$ vertices each. Both curves consist of one gadget per bit. These are either straight-line- or notch-gadgets, depending on the value of the respective bit. Thus, for $i \in \{1, \ldots, m\}$ we define the vertices of $\alpha$:

If $A[i] = 0$ then $v_{4i-3}^{\alpha} := (4i, 0)$, $v_{4i-2}^{\alpha} := (4i, 0)$, $v_{4i-1}^{\alpha} := (4i + 4, 0)$ and $v_{4i}^{\alpha} := (4i + 4, 0)$.
Otherwise $v_{4i-3}^{\alpha} := (4i, 0)$, $v_{4i-2}^{\alpha} := (4i, 1)$, $v_{4i-1}^{\alpha} := (4i + 4, 1)$ and $v_{4i}^{\alpha} := (4i + 4, 0)$.

And we define the vertices of $\beta$:

If $B[i] = 0$ then $v_{4i-3}^{\beta} := (4i, 0)$, $v_{4i-2}^{\beta} := (4i, 0)$, $v_{4i-1}^{\beta} := (4i + 4, 0)$ and $v_{4i}^{\beta} := (4i + 4, 0)$.
Otherwise $v_{4i-3}^{\beta} := (4i, 0)$, $v_{4i-2}^{\beta} := (4i, -1)$, $v_{4i-1}^{\beta} := (4i + 4, -1)$ and $v_{4i}^{\beta} := (4i + 4, 0)$.

We claim that

1. $\exists i \in \{1, \ldots, m\} : (A[i] = B[i] = 1) \Rightarrow d_F(\alpha, \beta) \geq 2$ and

2. $\forall i \in \{1, \ldots, m\} : (A[i] = 0 \vee B[i] = 0) \Rightarrow d_F(\alpha, \beta) < \sqrt{2}.$

To prove the first item, fix an arbitrary $i \in \{1, \ldots, m\}$. If $A[i] = B[i] = 1$, we have the vertices $v_{4i-3}^{\alpha} = (4i, 0), v_{4i-2}^{\alpha} = (4i, 1), v_{4i-1}^{\alpha} = (4i+4, 1)$ and $v_{4i}^{\alpha} = (4i+4, 0)$, as well as, $v_{4i-3}^{\beta} = (4i, 0),$ $v_{4i-2}^{\beta} = (4i, -1), v_{4i-1}^{\beta} = (4i+4, -1)$ and $v_{4i}^{\beta} = (4i+4, 0)$. Let $g \in \arg\inf_{h \in \mathcal{H}} \max_{t \in [0,1]} \|\alpha(t) - \beta(h(t))\|$. Now, assume that $d_F(\alpha, \beta) < 2$. This means, that $g$ must map $(4i + 2, 1) \in \overline{v_{4i-2}^{\alpha} v_{4i-1}^{\alpha}}$ to some point that lies closer than $(4i + 2, -1) \in \overline{v_{4i-2}^{\beta} v_{4i-1}^{\beta}}$. This is a contradiction, because the circle of radius 2 around $(4i + 2, 1)$ does only intersect one point of $\beta$, namely $(4i + 2, -1)$. In particular $v_{4i-3}^{\beta}$ and $v_{4i}^{\beta}$ have distance $\sqrt{5} > 2$.

To prove the second item, assume w.l.o.g. that $A[i] \neq B[i]$ for all $i \in \{1, \ldots, m\}$. Otherwise $\alpha$ and $\beta$ represent the same curve and have distance 0. Let $m = 1$ and w.l.o.g. assume that $B[1] = 1$. Then we have the vertices $v_1^{\alpha} = (4, 0), v_2^{\alpha} = (4, 0), v_3^{\alpha} = (4 + 4, 0)$ and $v_4^{\alpha} = (4 + 4, 0)$, as well as $v_1^{\beta} = (4, 0), v_2^{\beta} = (4, -1), v_3^{\beta} = (4 + 4, -1)$ and $v_4^{\beta} = (4 + 4, 0)$. Let $g$ be a reparameterization that maps $v_1^{\alpha}$ to $v_1^{\beta}$ and $v_4^{\alpha}$ to $v_4^{\beta}$, as well as $\overline{v_1^{\beta} v_2^{\beta}}$ and $\overline{v_3^{\beta} v_4^{\beta}}$ to some infinitesimally small sub-segment of $\overline{v_1^{\alpha} v_4^{\alpha}}$ each. Since these sub-segments have length less than 1 each, any point of these is mapped to a point within distance less than $\sqrt{2}$. Now, let $g$ map the remaining segment $\overline{v_2^{\beta} v_3^{\beta}}$ of $\beta$ linearly to the remaining middle sub-segment of $\overline{v_1^{\alpha} v_4^{\alpha}}$ of $\alpha$. Since this remaining sub-segment has length larger than 2, again any point is mapped to a point within distance less than $\sqrt{2}$. Since we can inductively apply this argument for any $m > 1$, i.e., any number of gadgets, we conclude that $d_F(\alpha, \beta) < \sqrt{2}$.

Now, suppose there exist oblivious randomized functions $S$ and $E$ not depending on the data such that $d_F(\alpha, \beta) \leq E(S(\alpha), \beta) \leq \eta \cdot d_F(\alpha, \beta)$ with constant probability, for an arbitrary $\eta \in [1, \sqrt{2}]$.

Alice computes the compressed representation $S(\alpha)$ using some of the public coins and communicates $S(\alpha)$ to Bob. Bob evaluates the estimator $E(S(\alpha), \beta)$.

If $E(S(\alpha), \beta) < 2$ then with constant probability $d_F(\alpha, \beta) \leq E(S(\alpha), \beta) < 2$.

If $E(S(\alpha), \beta) \geq 2$ then with constant probability $d_F(\alpha, \beta) \geq E(S(\alpha), \beta)/\sqrt{2} \geq \sqrt{2}$.

Thus, Bob can distinguish the above two cases and therefore solve the set disjointness problem with constant probability, which implies that $S(\alpha)$ consists of $\Omega(m)$ bits. $\qquad\square$