[Reviews · NeurIPS 2019]

Reviewer 1



The contribution to prove effectiveness of random projection for polygonal curves is novel and is a neat extension of previous work on median and center clustering of polygonal curves using Frechet distance. As I understand, the sublinear dependence for 1-median clustering does not have much novelty beyond using a previously known sampling idea from Indyk's thesis (2000), already cited by the authors. I found the empirical results meaningful and the authors do improve the efficiency of Alt-Godau implementation by an order of magnitude or more. However, I cannot judge the empricial work and its significance or broader impact effectively as I am familiar with the previous theoretical results on this problem but have not studied their practical applications carefully.

Reviewer 2



This paper demonstrates a theoretically guaranteed approximation for computing the Frechet distance between two polygonal curves.The paper also provides a CUDA-parallelized implementation of the Alt-Gadau algorithm for computing the distance (parallelizing the distance computation over each pair of line segments, one from each curve). In the experiments, the paper 1) Verifies the accuracy of the approximate JL distance computation 2) Demonstrates the speedup in runtime - 3 hours vs 40 seconds on data consisting of 6 curves in 6K dimensional with 2K points in each. From eyeballing the results (Fig 2), the JL projection and parallelization each contribute roughly an order of magnitude (10X) to the speedup. 3) Applies the algorithm to cluster real word data (based on k-center approximation algorithm by [Buchin 2019a]). Finally, it also proposes a k-centers approximation (following [Indyk 2000], ongoing work). The theoretical contribution of the paper is a rather straightforward application of the JL algorithm, by projecting each vertex of the curve and applying the JL distortion guarantees. The main technical contribution here is a distance computation lemma (Lemma 8). While source code is provided for the CUDA implementation, there is no mention of any algorithmic advances to parallelize the algorithm. In the experimental section, in 1) it would help to have a discussion of the theoretical gain in performance from the JL projection (presumably ~log d*eps^2/d factor) and the tradeoff of quality vs speed. In 2) the speed-up of 3 hours to 40 sec is substantial. However this speed up is a combination of the JL projection and the parallization. It would be useful to decouple the contributions of these two factors, and to also evaluate the runtime on a larger collection of datasets (beyond the single hydraulic test rig dataset). Finally, the validation of cluster quality in 3) is not very compelling. due to the absence of stated ground truth / lack of a clustering metric and a standard baseline algorithm for comparison. Using datasets (synthetic or otherwise) with well defined ground truth clusters, a choice of cluster quality metrics, and a baseline algorithm for comparison of cluster quality would improve this experiment. The paper is well organized. One section that could improve clarity is the description of the datasets which is rather long-winded. A table would be helpful in describing the dataset charateristics. The notion of filtering and heuristic filters in lines 244-255 was not clear. The plots in all figures can be improved (e.g. increasing font size in Fig 3, showing more clearly the speedups obtained in Fig 2). The number of clusters expected (line 265/266) for the DELTA dataset can be explicitly stated (I assume k=5). Addendum: I think that the authors do address the experimental concerns I originally had, specifically the point regarding comparison to baseline /clustering metrics. Regarding the theoretical results, in light of the other reviews, I agree that these contributions are of some significance. I am adjusting my score accordingly.

Reviewer 3



This paper provides theorems and experiments that the well-known curve distance, the (continuous) Frechet distance, is preserved under JL random projections. The main theoretical result is not just a (1+eps)-relative error, it also requires a roughly gamma * sqrt{eps} additive term where gamma is the maximum distance between two points. This is not too surprising to have the additive term, but also not entirely clear it is needed. The experiments use a state of the art code (from SOCG 2019) to compute the distance, and the paper mentions a GPU / parallel extension. It shows results where curves are of length n and the dimension d = Theta(n), that the random projection provides significant speed up with reasonable loss of accuracy. These results are nice, and the mathematical / theoretical part is very well written. I have few concerns: * the paper does not specify what dimension the curves are projected down to. It shows these results empirically in terms of eps. But the JL results these are based on are assyptotic results require O((1/eps)^2 * log (n/delta)) dimensions. The paper does not say what constant is assumed, what delta is, and what n is. I can only guess they are all 1, but am not sure. * It would be useful to compare the result to just using PCA, and also understanding when PCA works just as well as JL for these problems, empirically and theoretically. Often PCA works much better in practice, but can be slower. Does that hold in this setting? * I am not completely sold on the application of the Frechet distance in high dimensions. Is this a good modeling choice, and how is that justified? In particular, why focus on the bottleneck-distance based Frechet, and not DTW or something else that uses a more "average" case approach to curve distances. And do the sequence information really matter as opposed to the distribution of the n data points (and thus a cosine or KL sort of distance)? Overall, the paper is fairly nicely done, and the results will generate some interest, so I am ok accepting. But there are some (small) concerns with the presentation of the experiments, and with the empirical motivation (I'll add, mathematically, the question is clearly interesting). Typo in Theorem 11. I think "(2+delta)-app..." should "(2+eps)-app..." ADDENDUM: I have read the response, and it has definitely addressed the main concerns above. I'd appreciate even more discussion on the application, since I think these insights into why specifically the Frechet distance is useful is pretty interesting. Also, the poor performance in this case of PCA (in initial experiments) is surprising, and deserves further investigation, IMO. The other comments, if leading to changes in the paper, I feel will make it stronger. I am elevating my score.

[Author Response · NeurIPS 2019]

PS1 PS3

random simplices

PCA vs. JL

We thank the reviewers for their detailed comments and suggestions, which we have addressed below.

**(#1) novelty of sub-sampling**: Please note that Thm. 11 and 12 involve a *candidate* set and a *witness* set. While the existence of sublinear witness sets are immediate from Indyk's work, his candidate set is the entire input of size $n$. Our own arguments lead also to sublinear (even constant) *candidate sets*: In Thm 11 this is mainly by triangle inequality, but in Thm 12 it is a more complicated argument (sketched in the text and detailed in the supplement) assuming a natural distribution of distances around the true median. Meanwhile the experiments show outstanding performance.

**(#1)(#3) additive error**: We believe that the additive error is necessary. Put two curves $p = p_1 p_2, q = q_1 q_2$ with Frechet distance 1 whose points are vertices of a rectangle with $\|p_i - q_i\|_2 = 1, \|p_1 - p_2\|_2 = \|q_1 - q_2\|_2 = \gamma$ for *very large* $\gamma$. The above distances (esp. $\|p_i - q_i\|_2 = 1$) between points will be distorted by at most $(1 \pm \epsilon)$ so the Frechet distance will also be within $(1 \pm O(\epsilon))$. Insert another point $z$ in the middle of one curve. $z$ has distance $\approx \gamma/2$ to each other point for reasonably large $\gamma$. Now JL has its mass concentrated in the interval $(1 \pm \epsilon)$ but inspecting the concentration inequalities nearly half of this mass is between $(1 + \frac{\epsilon}{100})$ and $(1 + \epsilon)$. Thus with reasonably large probability the error will be $> \frac{\epsilon\gamma}{100}$ which is additive since $\gamma$ is unrelated to the original Fréchet distance of 1.

**(#2) abscence of stated ground truth / lack of clustering metric and baseline**: An evaluation of the meaningfulness of the $k$-center objective for a real world problem is out of scope of this work. Our starting point is: given that $k$-center gives meaningful results, how well does our approximation resemble those results? Thus, from our point of view, there is no abscence of ground truth. We ran the Buchin et al. $k$-center algorithm to obtain clusters. Those were confirmed to expose a meaningful structure in the data. This acts as our ground truth. Our focus, as stated in **Q3**, lies in a comparison of the algorithms quality with/without embedding. To answer **Q3** we wrote "In about 99.75% of the 400 experiments for the DELTA data set the same center-set was identified..." The exact clusters and centers are available in the supplement.

**(#2) decoupling of the contributions of the embedding and the parallelization**: In Fig. 2. the random projection and the parallelization speed up the computations by a factor of 10 *each independently*. *Both together* yield a speedup of factor 100. The requested *decoupling* is thus already presented. We will do our best to make his clear. Cf. left plot.

**(#2) theoretical gain and quality vs speed**: Indeed since the algorithms run in linear time with respect to the dimension $d$, the gain is a factor of $(4\epsilon^{-2} \log n)/d$. We will add this explicitly to the discussion. It is a good idea and we will add quality vs speed trade-off plots comparing to other baselines like PCA, see reviewer (#3) and right plot.

**(#2) further data sets**: We will simulate further data to outline the running time performance on high-dimensional data, e.g. from the $(d + 1)$-dimensional simplex for exceptionally large $d$, see middle plot.

**(#3) target dimensions and constants**: For the dimensions of the original data we will add a table, see minor comment of (#2). For the target dimensions we indeed use the asymptotic formula with a factor of 4, i.e., $k = 4\epsilon^{-2} \ln n$. We will add this and the resulting values of k for all experiments. An extensive empirical study [1] showed that a factor of 2 almost never fails. Another factor of 2 was added to be absolutely sure corresponding to $\delta = \frac{1}{4}$.

**(#3) Fréchet distance and sequence information**: Proposition 7 and Lemma 8 can be used to prove an error-bound for the continuous DTW, which we will add. However our focus is on a physical application that involves numerous sensors ($\approx$600, even more in the future). These sensors, though of same type, have varying noise and bias due to aging, radiation, calibration, and manufacturing variations. Our focus lies in finding outliers among time-sequences and thus need measures sensitive to outliers. Sum-based DTW is not applicable since it is sensitive to bias/noise and averages out single heavy outliers. The sequence information does matter, because the notion of an outlier is relative to the single curves noise/bias level. Further, Fréchet distance (already stated in our introduction) induces a linear interpolation. This interpolation is crucial because sensors are sometimes not available for short periods of time (due to radiation, heat etc.). Using a discrete distance measure like the discrete Fréchet or DTW, this could induce an unboundable additive error, which would require us to pre-process the data and search for sampling gaps and is not desirable.

**(#3) PCA**: Empirically, we have started evaluating the quality and running-time of the PCA vs. the random projection, see right plot. We will also try to obtain theoretical results, which is part of future work.

[1] Suresh Venkatasubramanian and Qiushi Wang. The johnson-lindenstrauss transform: An empirical study. In *Proceedings of ALENEX*, pages 164–173, 2011.

[Meta-Review · NeurIPS 2019]

The detailed and thoughtful author response effectively addressed many of the concerns of the reviewers, both on the importance of the theoretical problem, and the meaning of the empirical results. This pushed the paper over the threshold for acceptance.